



# Harmonization of Global Land-Use Change and Management for the Period 850-2100 (LUH2) for CMIP6

George C. Hurtt[1][*][φ], Louise Chini[1], Ritvik Sahajpal[1], Steve Frolking[2], Benjamin L. Bodirsky[3],
Katherine Calvin[4], Jonathan C. Doelman[5], Justin Fisk[1,6], S. Fujimori[7], Kees Klein Goldewijk[5,8],
Tomoko Hasegawa[7], Peter Havlik[9], Andreas Heinimann[10], Florian Humpenöder[3], Johan
Jungclaus[11], Jed Kaplan[12], Jennifer Kennedy[1], Tamas Krisztin[9], David Lawrence[13], Peter
Lawrence[13], Lei Ma[1], Ole Mertz[14], Julia Pongratz[11,15], Alexander Popp[3], Benjamin Poulter[16],
Keywan Riahi[9], Elena Shevliakova[17], Elke Stehfest[5], Peter Thornton[18], Francesco N. Tubiello[19],
Detlef P. van Vuuren[5,8], Xin Zhang[20]

*Correspondence to:* gchurtt@umd.edu

φ authors alphabetical after Frolking.

[1]Department of Geographical Sciences, University of Maryland, U.S.A.
[2]Institute for the Study of Earth, Oceans, and Space, University of New Hampshire, U.S.A.
[3]Potsdam Institute for Climate Impact Research, Germany
[4]Joint Global Change Research Institute, Pacific Northwest National Laboratory, U.S.A.
[5]PBL Netherlands Environmental Assessment Agency, Netherlands
[6]Applied Geosolutions, U.S.A.
[7]National Institute for Environmental Studies, Japan
[8]Copernicus Institute of Sustainable Development, University of Utrecht, Netherlands
[9]International Institute for Applied Systems Analysis, Austria
[10]Institute of Geography and Centre for Development and Environment, University of Bern,
Switzerland
[11]Max Planck Institute for Meterology, Hamburg, Germany
[12]Department of Earth Sciences, University of Hong Kong, Hong Kong
[13]National Center for Atmospheric Research, U.S.A.
[14]Department of Geosciences and Natural Resource Management, University of Copenhagen,
Denmark
[15]Ludwig-Maximiliansuniversität Munich, Department of Geography, Germany
[16]NASA Goddard Space Flight Center, Biospheric Sciences Lab, Greenbelt, MD 20771, U.S.A.
[17]Geophysical Fluid Dynamics Lab, Princeton, NJ, U.S.A.
[18]Oak Ridge National Laboratory, U.S.A.
[19]Statistics Division, Food and Agriculture Organization of the United Nations
[20]Appalachian Laboratory, University of Maryland Center for Environmental Science, U.S.A.





**Abstract**. Human land-use activities have resulted in large changes to the biogeochemical and biophysical properties of the Earth surface, with consequences for climate and other ecosystem services. In the future, land-use activities are likely to expand and/or intensify further to meet growing demands for food, fiber, and energy. As part of the World

Climate Research Program Coupled Model Intercomparison Project (CMIP6), the international community is developing the next generation of advanced Earth System Models (ESMs) to estimate the combined effects of human activities (e.g. land use and fossil fuel emissions) on the carbon-climate system. A new set of historical data based on the History of the Global Environment database (HYDE), and multiple alternative scenarios of the future (2015-2100) from Integrated Assessment Model (IAM) teams, are required as input for these models. Here we present results from

the Land-use Harmonization 2 (LUH2) project, with the goal to smoothly connect updated historical reconstructions of land-use with new future projections in the format required for ESMs. The harmonization strategy estimates the fractional land-use patterns, underlying land-use transitions, key agricultural management information, and resulting secondary lands annually, while minimizing the differences between the end of the historical reconstruction and IAM initial conditions and preserving changes depicted by the IAMs in the future. The new approach builds off a similar

effort from CMIP5, and is now provided at higher resolution (0.25 x 0.25 degree), over a longer time domain (850-2100, with extensions to 2300), with more detail (including multiple crop and pasture types and associated management practices), using more input datasets (including Landsat remote sensing data), updated algorithms (wood harvest and shifting cultivation), and is assessed via a new diagnostic package. The new LUH2 products contain >50 times the information content of the datasets used in CMIP5, and are designed to enable new and improved estimates

of the combined effects of land-use on the global carbon-climate system.


## 1. Introduction

Over the past several centuries to millennia, human land-use activities have grown and intensified to provided food, feed, energy, and fiber to support an expanding human population. These same land-use activities have also resulted in large changes to the underlying biogeophysical properties of the Earth surface, with impacts on climate, biogeochemical cycling, and habitat for biodiversity. In the future, land-use activities are likely to expand and/or intensify further to meet future demands for food, feed, energy, and fiber. What have been the effects of land-use activities on the climate system? What will be the impacts on climate of future land-use scenarios? Addressing these questions requires an integrated set of historical land-use data, integrated assessment models of the future, and climate models. To be most useful, requisite land-use data must be global, spatially and temporally and conceptually consistent from the past through to the future, and in a format that is usable by Earth System Models (ESMs).

Previously, in preparation for the Fifth Assessment Report (AR5) of the Intergovernmental Panel on Climate Change (IPCC) and as part of CMIP5, the Land-use Harmonization (LUH1) project provided harmonized land-use data for the years 1500-2100, at 0.5° x 0.5° resolution (Hurtt et al., 2011). These data served as required land-use forcing for CMIP5 climate model experiments and have been used in numerous related studies to assess the effects of land-use change on carbon and climate (Brovkin et al., 2013; Jones et al., 2011; Shevliakova et al., 2009; Shevliakova et al., 2013). They have also been extended for use in uncoupled DGVM modeling studies (e.g. TRENDY) and as input to the Global Carbon Project (Le Quéré et al., 2013; Le Quéré et al., 2014; Le Quéré et al., 2015; Le Quéré et al. ,2016; Le Quéré et al., 2017; Le Quéré et al., 2018; Friedlingstein et al., 2019) and other studies (Jones et al., 2013; Di Vittorio et al., 2014; Collins et al., 2015; Thornton et al., 2017; Di Vittorio et al., 2018)

Now, as part of the World Climate Research Program Coupled Model Intercomparison Project (CMIP6, Eyring et al., 2016), the international research community has developed the next generation of advanced ESMs able to estimate the combined effects of human activities (e.g. land use and fossil fuel emissions) on the carbon-climate system. In addition, a set of historical data based on the History of the Global Environment database (HYDE), and multiple alternative scenarios of the future (2015-2100), developed by Integrated Assessment Model (IAM) teams, have been developed as drivers for these models. The goal of the Land-Use Harmonization (LUH2) project is to prepare a new harmonized set of land-use scenarios that smoothly connects the historical reconstructions of land-use with the future projections in the format required for ESMs. The land-use harmonization strategy estimates the fractional land-use patterns, underlying land-use transitions, and key agricultural management information, annually for the time period 850-2100 at 0.25° x 0.25° resolution, while minimizing the differences at the transition between the historical reconstruction ending conditions and IAM initial conditions, and working to preserve changes depicted by the IAMs in the future. The resulting data products are a required input for multiple CMIP6 model experiments, including the historical all-forcing experiment, and related model intercomparison project



experiments including PaleoMIP (Junclaus et al., 2017), ScenarioMIP (O'Neill et al., 2016), LUMIP (Lawrence et al., 2016). Extensions are also provided for 2100-2300 as input to climate stabilization experiments.

## 2. Methods

Like its predecessors, The Global Land Use Model (Hurtt et al., 2006; Hurtt et al., 2011), GLM2 computes subgrid-scale land-use states and corresponding transition rates using an accounting-based method that tracks the fractional state of the land surface in each grid cell as a function of the land surface at the previous time step, and a transition matrix. This can be represented using the following matrix equation:

$$l(x,t+1) = A(x,t)l(x,t)$$

$$x = (1,…,N), \ t = (t0,…,tf) \tag{1}$$

where $l(x,t)$ is a vector giving the fractions of grid cell area in each land-use category in a grid cell $x$ and time $t$, and $A(x,t)$ is a matrix giving the land-use transition rates between N land-use categories in grid cell $x$ and time $t$. Each

element, $a_{ij}(x,t)$ of the matrix $A(x,t)$ gives the rate at which land-use type $j$ was converted to land-use type $i$ between $t$ and $t+1$.

$$A(x,t) = \begin{bmatrix} a_{11}(x,t) & \cdots & a_{1n}(x,t) \\ \vdots & a_{ij}(x,t) & \vdots \\ a_{n1}(x,t) & \cdots & a_{nn}(x,t) \end{bmatrix} \ (i,j=1...N) \tag{2}$$

GLM2 was adapted and extended from GLM1 to track a larger list of 12 subgrid scale land-use types (i.e. 4 "natural land" types, 5 crop types, 2 pasture types, and urban), and key management information (i.e. fraction irrigated, fraction flooded, fraction biofuel, and rate of industrial N fertilizer application) related to agriculture,

$$m(x,t) = f(x,t) \tag{3}$$


where $m(x,t)$ is a vector of cropland management information for grid cell $x$ at time $t$, and $f(x,t)$ represents the functional dependence of management activities on location and time.

GLM2 was used to solve Eq. 1 and associated values of $A(x,t)$ and $m(x,t)$ annually for every 0.25° x 0.25° terrestrial

grid cell globally for 850-2100 (with extensions to 2300). In the process, the framework was used to determine on the order of $10^{10}$ unknowns. Since this was a large and underdetermined system, the approach was to solve the system for every grid cell at each time step by constraining with inputs including: (i) land-use maps, (ii) crop type





and rotation rates, (iii) shifting cultivation rates, (iv) agriculture management, (v) wood harvest, (vi) forest transitions, and (vii) potential biomass and biomass recovery rates. Because these inputs do not uniquely constrain

the system, additional assumptions were made including: (viii) the priority of primary (not harvested, cut or converted since 850 CE) or secondary land for wood harvesting and agricultural conversion, (ix) the inclusiveness in wood harvest statistics of wood cut in conversion of forest to agricultural use, and (x) the spatial pattern of wood harvest. Model input-output is illustrated in Fig. 1, and described below.

### 2.1 Historical Maps of Land Use

Historical maps of land use were based on the History of the Global Environment database (HYDE). HYDE provides long-term historical, spatially-explicit time series on a 5 arc minute resolution of population estimates as well as land use reconstructions covering the Holocene period, defined here as 10 000 BCE until the present (Table 1). It is an effort to quantify the agricultural expansion of humankind over time. In principle, HYDE uses a simple approach of combining historical population estimates with assumptions on the trajectory of historical land use per

capita. Allocation of land use patterns is steered at present day by satellite information and UN FAO agricultural data, and this is gradually replaced towards the past by a combination of spatially explicit maps such as climate, soil, slope, and neighborhood of rivers and lakes. The latest version (3.2; Klein Goldewijk et al., 2017) presents land use categories such as built-up area, managed pastures and more extensive rangelands, cropland excluding rice, and rice as a separate crop because of its relevancy for greenhouse gas emissions. A distinction was made between irrigated

and rain-fed cropland (both for other crops and rice). Besides the baseline reconstruction, two alternative historical land-use reconstructions were provided based on uncertainties. For a full description of the methodology see Klein Goldewijk et al. (2017).

The version of the HYDE 3.2 dataset used for the baseline LUH2 historical product was the "2016_beta_release" version, and the version used for the upper and lower scenarios was the "2017_beta_release_000" version. Data was

provided at 5' spatial resolution, every 100 years from 800 to 1700, every 10 years from 1700 to 2000, and then annually from 2000 to 2015. These data were aggregated to 0.25°×0.25° resolution and converted from absolute area of each grid cell to grid cell fractional area. Data were then linearly interpolated in time to produce annual maps of the fraction of each 0.25° grid-cell occupied by each of the following land-use types: cropland, grazing land, pasture, and urban. The ice and water fractions of each grid cell were also taken from the HYDE dataset and were assumed

constant over time. By subtracting the land-use and ice and water fractions from each grid cell, the fractions of each grid cell occupied by natural vegetation (either primary or secondary forest or grassland) were also determined. The HYDE 3.2 dataset also includes a global map that assigns a country code to each terrestrial grid cell, at 5' resolution. This map served as a basis to generate a similar map at 0.25° resolution, consistent with the 0.25° maps of land-use data. In this map every grid cell with ice/water fraction less than 1.0 was assigned a country code, resulting in a

global map containing 199 countries.



### 2.2 Historical Maps of Crop types and Crop Rotations

The cropland fraction of each grid-cell, along with transitions to/from cropland, are further sub-divided into five different crop functional types (CFTs): C3 annuals, C4 annuals, C3 perennials, C4 perennials, and C3 nitrogen fixers. For the years 850 to 2015 the CFT fractions of total cropland are primarily based on data from Monfreda et al. (2008), which provides global maps of harvested areas of 175 different crops, at 5-minute spatial resolution, for the year 2000. For use in the LUH2 methodology, these maps were aggregated into five CFT classes at 0.25° spatial resolution and then normalized so that all CFT fractions sum to 1 in each grid-cell. For grid cells that do not have crop-type data from Monfreda et al., national crop-type data from FAO (FAO 2016) is used instead (i.e. by aggregating the 169 FAO crop types into the 5 CFT classes represented in LUH, averaging over all years of FAO data from 1961 to 2013, then assigning the normalized national CFT fractions to any grid-cells within each country that did not have Monfreda data). The resulting map of CFT fractions is used for all years 850-2015 to sub-divide the gridded cropland fraction and cropland-related transitions into CFT fractions and CFT-related transitions, by multiplying the cropland fraction of each grid-cell (and the cropland-related transitions to/from each grid-cell) by the CFT fractions map. Note that this process includes the inherent assumption that the fraction of a grid cell that was harvested for a crop type (i.e. the Mondreda et al. data) was roughly correlated with the fraction of the total cropland area that was occupied by that crop type.

For the years 2015-2100, we first identify one or two CFTs in the IAM data that have the greatest global area increase over the 85-year period. We then attempt to follow the gridded changes in fraction of cropland occupied by those CFTs, by first assigning as much of the cropland expansion transitions as possible to the expansion of those one or two CFTs, and then, when needed, by adding transitions between CFTs to re-assign area from CFTs with lower rates of increase (or even reductions) of area in the IAM data to the CFTs with large global increases in area. The result of this process is typically that the global area changes of CFTs in LUH2 tend to follow global area changes of CFTs in the IAM data, not just for the CFTs with the largest area changes, but for others as well. When there were no CFTs with significant changes over the 2015-2100 period, the contemporary CFT ratios were used to disaggregate total cropland area into CFT fractions for all years 2015-2100.

Crop rotations or the practice of growing a sequence of crops on an agricultural field, within or across growing seasons, is a key component of agricultural management, and has impacts on overall crop yields, nutrient cycling, fertilizer and water usage, water quality and biodiversity (Bullock, 1992). An example of such a crop rotation is the corn-soybean-corn rotation practiced extensively in the U.S. Midwest. We generated a national scale crop rotation dataset for the U.S to quantify rates of transition from one crop functional type to another and applied those rates to the crop functional types in LUH2. We use the USDA Cropland Data Layer (CDL, Sahajpal et al., 2014) to quantify unique crop rotations for U.S from 2012 – 2014 (Sahajpal et al., 2014). Assuming a crop rotation span of 3 years, and nearly 100 unique crops in the CDL, we could potentially have $10^6$ unique crop rotations. Empirically, there are close to 100,000 unique crop rotations in the U.S for that time-period. However, by aggregating different crop types to the crop functional types in LUH2 and merging similar rotations, we estimated transition rates between different



crop functional types in LUH2 and applied them after all other transitions between land-use types have been computed.

**2.3 Historical Data on Agriculture Management Activities**

Historical information on crop management activities included data on irrigation, flooded agriculture, and industrial
nitrogen fertilizer application rates. Data on irrigated area, and area of flooded rice, were obtained from HYDE. The
irrigated fraction of each crop type was computed during the historical period by dividing the HYDE 3.2 irrigated
fraction of each grid-cell by the HYDE 3.2 cropland fraction of each grid-cell. This fraction is then used as the
irrigated fraction of each crop sub-type. The fraction of C3 annuals that are flooded for rice is computed historically
by dividing the HYDE 3.2 flooded fraction of each grid-cell by the C3 annual fraction of each grid-cell. For
industrial nitrogen fertilizers, we used a recent global compilation of N fertilizer use for 1961-2011 (Zhang et al.,
2015) as our base data set. Countries without fertilizer data reported in Zhang et al. (2015) were assigned regional
mean values, based on the regional grouping of countries defined in Zhang et al. (2015). Fertilizer use between 1915
and 1960 was hindcast using global synthetic N fertilizer use totals from Smil (2001), and was forecast from 2012 to
2015 using an estimate of global industrial N fertilizer use based on data from the International Fertilizer
Association (IFA, 2015).  Decadal mean N-fertilizer rates by crop and country were computed from the Zhang et al.
(2015) data and were assigned to mid-decade year (e.g., the 1961-1970 mean was assigned to 1965).  To generate
country fertilizer application rates for 2015, which we did not compute as a decadal mean, we assumed that the
fertilization rate since 2005 has changed with a same scaling factor across all countries and crop types (as in Zhang
et al., 2015). Using the harvested area in 2015 from HYDE 3.2 (see Section 2.1), the fertilization rate for country $j$
and crop $k$ in 2015 is determined by

$$R_{j,k,2015} = R_{j,k,2005} \cdot (F_{2015,IFA}/A_{2015}) \div (F_{2005}/A_{2005}),$$

where $R_{j,k,t}$ is the N-fertilization rate by crop type ($j$) by country ($k$) by year ($t$) [kg N ha$^{-1}$ y$^{-1}$], and $A_t$ is the global
total crop area in year $t$ from HYDE 3.2, $F_{2015,IFA}$ is the global N fertilizer application in 2015, estimated by applying
the trend in 2006-2012 from the IFA data to extrapolate to 2015 from 2012, yielding $F_{2015,IFA}$ = 115 Tg N y$^{-1}$, and
$F_{2005}$ is the global total N fertilizer application estimated as the product of N fertilizer application rate in 2005
computed from Zhang et al. (2015) and LUH2 cropland area ($F_{2005}$ = 94 Tg N, the mean of 2001-2010, as above).

Fertilizer application rates were hindcast from the 1960s to rates for 1950, 1930, and 1915. Synthetic N fertilizer
rates in 1915 are set to 0.0 kg N km$^{-2}$ for all countries and crop types, as this was when the Haber-Bosch industrial
process was invented. Using global N consumption data from Smil (2001) for 1950 ($F_{1950,Smil}$ = 3.7 Tg N y$^{-1}$) and
1930  ($F_{1930,Smil}$ = 1.0 Tg N y$^{-1}$), and crop area from LUH2 ($A_{j,k,t}$, see Section 2.1), the synthetic N rates by crop and
country ($R_{j,k,t}$) were estimated for 1950, 1930, and 1915 as follows

$$R_{j,k,1950} = R_{j,k,1965} \cdot (F_{1950,Smil}) \div \Sigma[R_{j,k,1965} \cdot A_{j,k,1950}],$$



$$R_{j,k,1930} = R_{j,k,1965} \cdot (F_{1930,Smil}) \div \Sigma[R_{j,k,1965} \cdot A_{j,k,1930}],$$

$$R_{j,k,1915} = 0.$$

where the sum is over all countries ($j$ index) and crops ($k$ index). Finally, we generated annual synthetic N fertilizer rate values by country and crop functional type and year ($R_{j,k,t}$) by linearly interpolating between values for 1915, 1930, 1950, 1965, 1975, 1985, 1995, 2005, 2015.

### 2.4 Rates of Shifting Cultivation

We considered shifting cultivation to be a specific land use sequence of clearing, agricultural use typically for one to
several years, and subsequent abandonment of land to forest (or other natural vegetation) regeneration for three years to several decades ('fallow'). While likely widespread in the early millennia of agriculture (Olofsson & Hickler, 2007), more recently it has been restricted to the tropics (Ruthenberg, 1980). We use the recent analysis of the past, present, and future extent of shifting cultivation (Heinimann et al., 2017) to constrain its occurrence in LUH2. Heinimann et al. (2017) based their analysis on the early global map of the distribution of 'primitive
subsistence agriculture' (Butler 1980), a visual inspection of the distribution of shifting cultivation based on the 2000-2014 Global Forest Change (GFC) data set (Hansen et al., 2013) coupled with high-resolution satellite imagery, and an extensive expert survey on regional trends in shifting cultivation, querying lead authors of scientific publications on shifting cultivation over the past decade (Heinimann et al., 2017).

Heinimann et al. (2017) estimated the current area under shifting cultivation (cultivated + fallow) to be about 280
Mha, distributed extensively and heterogeneously across Central and tropical South America, tropical Africa, and tropical Southeast Asia (see Fig. 5 in Heinimann et al., 2017). For each 1x1° grid cell with detected signs of shifting cultivation, they also estimated its level of occurrence, including both active cropland and fallows, aggregated into five classes of the total land area in each grid cell: none (<1%), very low (1-9%), low (10-19%), moderate (20-39%) or high (≥40%). They project significant declines in shifting cultivation extent through the 21[st] century, with losses
by the end of the century of more than 80% in Africa and Latin America, and 100% in Asia, and extent at 1x1° in remaining areas to be low or very low (see Fig. 7 in Heinimann et al., 2017).

We created annual LUH2 shifting cultivation maps by linearly interpolating between the assumed shifting cultivation rates in 1850 and the expert opinion-based rates of 2010 (Heinimann et al., 2017). The 1850 shifting cultivation rates were assumed to fall in the 'high' category of 70%. The future shifting cultivation rates were
similarly computed by linearly interpolating between the 2010 and the assumed 2100 rates from the expert opinion survey of Heinimann et al., 2017. For LUH2, shifting cultivation involved cropland only (grazing land was included as part of shifting cultivation in LUH1 but not in LUH2). For all grid cells, we used the mid-range of shifting cultivation occurrence (e.g., 5% for 'very low', 15% for 'low', 30% for 'moderate', and 70% for 'high'), and assumed that these fractions also applied to the fraction of cropland involved in shifting cultivation. We also
assumed that the residence time for a patch of cropland involved in shifting cultivation was only 1 year. At each




time-step in our model, we then abandoned the Heinimann et al. (2017) prescribed percentage of total cropland area in the grid cell (e.g. cropland to secondary land), and cleared the same area from natural vegetation (e.g. forest to cropland), with a prioritization of clearing secondary land first unless the available secondary land was less than 10 times the cropland area involved in shifting cultivation (based on an assumption of a 10-year fallow period). The global area of shifting cultivation activity tends to track global changes in cropland area from HYDE 3.2 (Klein Goldewijk et al., 2017, or see Section 2.1), and global future cropland area changes from IAMs, although this relationship between cropland area and shifting cultivation area declines over time due to the extent of shifting cultivation declining significantly, especially through the 21st century.

**2.5 Historical Statistics on Wood Harvest**

Historical wood harvest in LUH2 is based on national statistics, and partitioned into fuelwood and non-fuelwood, for 199 countries, based on a 1990 country list from HYDE 3.2 (Klein Goldewijk et al., 2017). For the years 1961-2015 the LUH2 wood harvest data is based on FAO national wood harvest volume data (FAO 2016) for both coniferous and non-coniferous round wood, which is combined with wood density values of 0.225 Mg C m$^{-3}$ for coniferous wood and 0.325 Mg C m$^{-3}$ for non-coniferous wood (Houghton and Hackler, 2000) to convert volume statistics to mass of carbon harvested. Harvest rates were hindcast to 1920 by interpolating from mean FAO per capita harvest rates from 1961-1965, using national population totals from HYDE 3.2 (see section 2.1), and national per capita fuelwood ('firewood') and timber ('sawtimber') wood harvest totals from 1920 (Zon and Sparhawk 1923). Note that Zon and Sparhawk totals for timber consumption include volume of wood for construction, industry, and pulp, and so, with firewood, should be roughly comparable to FAO 'total roundwood'.

For the years prior to 1920, national annual per capita wood harvest rates were computed in three different ways for low, baseline, and high LUH2 scenarios, and use the same national population data from HYDE 3.2 to compute the total national wood harvest (in Mg C) per year for each scenario. For the "low" wood harvest scenario, the national annual per capita wood harvest rates from Zon and Sparhawk (1923) were held constant for all years from 850 to 1920. However, prior to the fossil fuel era, global mean per capita wood harvest was likely significantly higher than in 1920, so for the "high" scenario we used a national per capita wood harvest demand reconstruction for "fuelwood" and "durable wood" from Kaplan et al. (2017) for the period 850-1800. Per capita wood harvest rates then transitioned linearly from 1800 rates to the 1920 rates of Zon and Sparhawk (1923), to mimic the global shift in energy sources from biomass towards fossil fuels (Smil, 2003). These high and low wood harvest scenarios represented two different extremes in terms of cumulative wood harvested and total area of forests removed. In addition, the high scenario is significantly higher than the LUH1 wood harvest reconstruction. To provide a scenario somewhere between these two extremes, we also generated a "baseline" wood harvest scenario in which we modified the Kaplan national wood harvest rates from 850 to 1800 by national scale factors. These scale factors are defined as twice the contemporary FAO national per capita wood harvest rates divided by the national per capita wood harvest rates in 1800 from the Kaplan data, and this definition was determined from analysis of the global time-series figure of historical biofuels consumption (Smil 2003) which shows current global per capita biofuels





consumption of around 6 GJ per capita and around 21 GJ per capita in 1800. Reducing the Kaplan wood harvest rates via these scale factors does not imply that the original Kaplan rates are too high, rather that the Kaplan data is likely to be capturing types of wood harvest and related processes that our model does not currently simulate. For years between 1800 and 1920 we linearly interpolate between the modified year 1800 rates from Kaplan and the Zon

and Sparhawk (1923) rates in 1920.

For the "low" and "baseline" scenarios, the reconstructed national wood harvest data were increased by a slash fraction of 30% (as in LUH1, Hurtt et al., 2011) to account for non-harvested losses from forests that occur during the wood harvesting process. For the "high" scenario, we do not add a slash fraction to the data for the years 850-1800 since it is assumed this is already included in the Kaplan data (Kaplan et al. 2017). In this scenario, the slash

fraction is linearly increased from 0% to 30% during 1800 to 1920, and held constant thereafter.

All national wood harvest totals from FAO and Zon and Sparhawk are assumed to represent the amount of wood produced by each country. In contrast, the data from Kaplan represents the wood harvest demand from each country, although it is assumed that during the years 850-1800 there was limited wood trade in most parts of the world, and hence demand would equal production. In Europe, however, international wood trade occurred during 850-1800

(Kaplan et al., 2017). So, for European countries only, if the available national biomass is not sufficient to meet the national wood harvest demand in a particular year, we seek the unmet demand from other European countries (i.e. increase the wood harvest production in other countries) proportional to the available biomass in each country. From 1500–2005, the global cumulative total wood harvest in the baseline scenario was 190 Pg C including slash (Fig. 2), compared with 142 Pg C and 381 Pg C in the "low" and "high" scenarios, respectively.

**2.6 Historical Maps of Forest Transitions**

The spatial patterns of forest transitions, particularly those related to wood harvesting, were constrained by the Landsat-based gridded forest loss observations from Hansen et al. (2013). This product consists of global 30m grids of tree canopy cover for year 2000 and gross forest cover loss and gain for the 2000-2012 time interval mapped using the entire global Landsat data archive (although only the forest loss data was used within LUH2). Within this

dataset, forest was defined using a single tree canopy cover threshold to match the global forest extent provided by the FAO FRA report (FAO 2000). Cumulative forest area was estimated by summing pixels with different tree canopy cover. Then the threshold was selected that most closely enabled a match to the total world forest cover for year 2000, which is 4085 million ha, according to FAO data. A threshold of 28% tree canopy cover produced 100.5% of the FAO forest area. This threshold was used to define forest area for the year 2000 at 30m spatial

resolution. Gross forest cover loss was reported only within areas covered with forest in the year 2000. Gross forest cover gain was mapped independently outside areas forested in the year 2000 and represents gain of tree canopy cover to 30% or higher from non-forest state. The global maps of forest extent and change were then aggregated to the same spatial resolution and format as the LUH1 datasets (0.5° × 0.5° fractional). To aggregate the data to the 0.5° grid, the area of each class was computed within each grid cell, and then the class area percent of total cell area

was calculated. The 0.5° product shows percent forest cover for year 2000 and percent gross forest cover loss and



gain during the 2000-2012 time interval. The 0.5° product was later downscaled to 0.25° for consistency with the new LUH2 spatial resolution. A very simple downscaling method was employed that kept the fraction of forest area (or forest loss) equal within each 0.25° grid-cell inside the 0.5° grid-cell cells.

The resulting map of forest loss was used within LUH2 as part of the algorithm for determining the spatial pattern of forest loss from wood harvesting. However, it should be noted that the Landsat-based forest loss maps differ from the LUH2 forest loss maps in multiple ways, including definitions of "forest" (i.e. tree canopy cover vs. biomass density), whether or not a single grid-cell can contain both forest and non-forest (LUH2 grid-cells are either potentially forested or potentially non-forested), whether or not the forest loss includes natural disturbances such as fires or not (LUH2 forest loss results only from land-use-related changes). As a result, the match between these
products is not perfect, and the Landsat-based forest loss data is used as a guide to improving the LUH2 forest loss patterns, rather than a hard constraint on those patterns.

### 2.7 Biomass Density and Recovery Rates

To discriminate forested land from non-forested land, and to convert quantities of harvested wood in biomass units into harvested area, information was needed on the historical distribution of forests and above ground carbon stocks.
As no complete global, gridded, historical record of these quantities was available, a simple empirically-based global terrestrial model was used to provide a consistent set of both global forest cover and carbon stocks. Estimates of ecosystem properties were based on an updated version of the MIAMI-LU ecosystem model (Hurtt et al., 2002; Hurtt et al., 2006; Hurtt et al., 2011). Miami-LU was driven by the empirically-based Miami model of net primary production (Leith, 1972), which has integrated sub-models of plant mortality and disturbance. The model tracked
sub-grid heterogeneity resulting from land-use changes in a manner similar to the more advanced Ecosystem Demography (ED) model (Hurtt et al., 1998; Moorcroft et al, 2001; Hurtt et al., 2002).

Miami-LU was run globally at 0.5° x 0.5° resolution for a spin-up period of 500 years using data from the Multi-Scale Synthesis and Terrestrial Model Intercomparison Project (MsTMIP) (Wei et al., 2013). These data are a combination of climatologies from the Climate Research Unit and National Centers for Environmental Protection,
and has a global 0.5° x 0.5° climatology with a 6 hourly daily time step from 1901 – 2010. MIAMI-LU outputs were subsequently downscaled to 0.25° x 0.25° resolution to match the remaining LUH2 inputs (downscaling simply assigned all 0.25° x 0.25° grid-cells the same fraction value as the 0.5° x 0.5° grid-cell they were contained within). Aggregated globally, the NPP estimate from Miami-LU was 63 Pg C y$^{-1}$. This fell within a range of NPP estimates from various global biogeochemical models, ranging from 40 Pg C y$^{-1}$ to 81 Pg C y$^{-1}$ (Cramer et al. 1999).
Miami-LU estimated a global stock of potential plant carbon of 718 Pg C (Figure 3). This fells within a range spanning 557 Pg C (Kucharik et al., 2000) to 923 Pg C (Sitch et al., 2003), with a more recent estimate of 772 Pg C (Pan et al., 2013). The total potential above-ground carbon stock was 563 Pg C. To differentiate forest from non-forest areas, a definition based on potential above-ground standing stock of 2 kg C m$^{-2}$ was used (Hurtt et al., 2002; Hurtt et al., 2006; Hurtt et al., 2011). Each grid cell was thus identified as potential forest or potential non-forest



based on potential biomass.   Using this definition, 48.8 x 10$^6$ km$^2$ of the land surface was classified as potential
forest. For comparison, potential forest area based on the BIOME model was estimated at 60 x 10$^6$ km$^2$ (Klein
Goldewijk, 2001). Finally, Miami-LU was also used to estimate the recovery of carbon stocks on secondary lands by
tracking the mean age of secondary land in each grid cell, although not explicitly account for the full age distribution
or the potential effects of land degradation, management, or pollution that may have occurred.


### 2.8  Future Land Use, Wood Harvest, and Management from Integrated Assessment Models

For 2015-2100, land use and wood harvest information were based on eight different marker SSP-RCP scenarios
derived from five different Integrated Assessment Models. These data sets were prioritized as input to CMIP6
climate model simulations by ScenarioMIP, are fully described elsewhere (O'Neill et al., 2016), and are summarized
below and in Table 2 in the order described in O'Neill et al. (2016).

#### 2.8.1     SSP5-8.5 REMIND-MAGPIE

The scenario SSP5-8.5 is based on the REMIND-MAgPIE SSP5 baseline scenario, which has a radiative forcing
close to RCP8.5 (Kriegler et al., 2017). SSP5 is characterized by rapid and resource intensive development and
material-intensive consumption patterns, whereas technological progress, including agricultural productivity, is
high. In consequence, the SSP5-RCP8.5 scenario exhibits very high levels of fossil fuel use, up to a doubling of
global food demand, and up to a tripling of greenhouse gas emissions over the course of the century, marking the
upper end of the emission scenario literature. The REMIND-MAgPIE integrated assessment modeling framework
consists of the Regionalized Model of Investment and Development (REMIND) and the Model of Agricultural
Production and its Impacts on the Environment (MAgPIE). REMIND (Luderer et al., 2015) is a global multi-
regional energy-economy general equilibrium model linking a macro-economic growth model with a bottom-up
engineering-based energy model. MAgPIE (Popp et al., 2014) is a global multi-regional partial equilibrium model of
the land-use sector, which accounts for spatially explicit biophysical constraints derived by the vegetation,
hydrology and crop growth model LPJmL (Müller and Robertson, 2014; Bondeau et al., 2007; Bodirsky et al.,
2012). Land-use decisions in MAgPIE are modeled at a spatially-explicit level (Lotze-Campen et al., 2008).
REMIND and MAgPIE are coupled by exchange of price and quantity information on bioenergy and GHG
emissions (Popp et al., 2011; Kriegler et al., 2017). As an outcome of the strongly increasing food and feed demand
as well as highly intensified future livestock production systems relying on concentrates rather than roughage feed
(Weindl et al., 2017), the SSP5-RCP8.5 scenario shows strong expansion of global cropland into pasture and forest
land, with an increase of about 300 Mha (20%) between 2010 to 2100.

#### 2.8.2     SSP3-7 AIM

The SSP3-7.0 is a simulation derived from the SSP3 baseline scenario (Fujimori et al., 2017) which has a radiative
forcing close to 7.0 Wm$^{-2}$. The SSP3-7.0 was simulated using the Asia-Pacific Integrated assessment



Model/Computable General Equilibrium model (AIM/CGE; (Fujimori et al., 2014; Fujimori et al., 2012)) combined with a land-use allocation model (Hasegawa et al., 2017). AIM/CGE is a global integrated assessment model, coupling representations of economy, energy systems, land, and climate. AIM/CGE is a recursive dynamic general equilibrium model, adjusting prices until the supply and demand for energy, industrial, agriculture, forest commodities as well as all the other goods and services equilibrate. AIM/CGE includes 17 regions and 42 industrial classifications including 10 agricultural sectors. The land system is divided into nine agro-ecological zones. Land use and land cover were further downscaled to 0.5 x 0.5 grids using the land allocation approach developed by Hasegawa et al. (2017). SSP3 is a world of regional rivalry where countries increasingly focus on domestic and regional issues. Economic development is slow, consumption is material-intensive, and population growth is low in industrialized and high in developing countries. Land use change is hardly regulated. Agricultural land intensification is low, especially due to very limited transfer of new agricultural technologies to developing countries. Unhealthy diets with high animal shares and high food waste prevail. A regionalized world leads to reduced trade flows for agricultural goods. The SSP3-RCP7.0 scenario includes strong expansion of global crop and pasture land, with increases of 40% and 7% from 2010 to 2100, respectively, resulting in large-scale deforestation.

### 2.8.3    SSP2-4.5 MESSAGE

SSP2-4.5 is a low stabilization scenario that stabilizes radiative forcing at 4.5 Wm$^{-2}$ (~650 ppm CO2-equivalent) before 2100 without ever exceeding that value. RCP4.5 is simulated in a structure of interlinked disciplinary and sectorial models referred to as the IIASA Integrated Assessment Modelling (IAM) framework (Riahi et al. 2007, Fricko et al. 2017). Within the framework, land-use dynamics are modelled with the GLOBIOM model, which is a recursive-dynamic partial-equilibrium model (Havlík et al., 2011). GLOBIOM includes a bottom-up representation of the agricultural, forestry and bio-energy sector, which allows for the inclusion of detailed grid-cell information on biophysical constraints and technological costs, as well as a rich set of environmental parameters, including comprehensive AFOLU (agriculture, forestry and other land use) GHG emission accounts and irrigation water use. For spatially explicit projections of the change in afforestation, deforestation, forest management, and their related CO2 emissions, GLOBIOM is coupled with the G4M model (Kindermann et al., 2006; Kindermann et al., 2008; Gusti, 2010). These models are linked to the MESSAGE energy system model (Messner and Strubegger, 1995; Riahi et al., 2012), while air pollution implications are derived with the help of the GAINS model. An important feature of the RCP4.5 is the initial decrease in forest by about 43 million ha from 2000 to 2050 (comparable to the reference scenario), with a subsequent increase in forest by about 331 million ha from 2050 to 2100.

### 2.8.4    SSP1-2.6 IMAGE

The SSP1-2.6 scenario is developed using the IMAGE 3.0 integrated assessment model (Stehfest et al., 2014). IMAGE is a model framework describing the future agriculture system and energy system, as well the changes in future land cover, the carbon and hydrological cycle and climate change. While most socio-economic processes are described at the level of 26 regions, environmental processes are modeled on a grid -basis (30 or 5 arc-minutes). The LPJmL model





is hard-coupled to IMAGE on a yearly basis (Mueller et al., 2016), and calculates for crops & grassland productivity, natural vegetation dynamics, hydrology, and the carbon cycle. The SSP1-RCP2.6 is derived from the SSP1 baseline scenario which projects a future under a green growth paradigm (van Vuuren et al, 2017). The SSP1 scenario is

characterized by moderate population growth leveling off by mid-century, and by high economic growth and technological improvements including agricultural productivity. In addition, SSP1 describes an environmentally aware world concerned with limiting biodiversity loss and reduced appetite for animal product consumption. Mitigation policy is added to the SSP1 baseline scenario to achieve a maximum warming of 2 degrees consistent with the RCP2.6 scenario (van Vuuren et al., 2011). Important policies from the land-use perspective are increased bio-

energy use in combination with carbon capture and storage, avoided deforestation policy to reduce deforestation, and restoration of degraded forests (Doelman et al., 2018).

In SSP1-2.6, the combination of socio-economic trends and climate policy results in substantial reductions in total agricultural land. At the same time, large areas are dedicated to bioenergy production, and also forest area increases (Doelman et al., 2018; Popp et al., 2017).

**2.8.5    SSP4-6.0 GCAM**

The SSP4-6.0 is a simulation derived from the SSP4 baseline (Calvin et al., 2017), with a modest climate policy imposed to limit 2100 radiative forcing to 6.0 $Wm^{-2}$. The SSP4-6.0 was simulated using the Global Change Assessment Model (GCAM; Wise et al., 2014). GCAM is a global integrated assessment model, coupling representations of energy, water, land, economy, and climate. GCAM is a market-equilibrium model, adjusting

prices until the supply and demand for energy, agriculture, and forest commodities equilibrate. GCAM subdivides the world into 32 economic regions. The land system is further subdivided into as many as 18 agro-ecological zones, resulting in 283 agriculture and land use regions. Land use and land cover were further downscaled to 0.5° x 0.5° grids using the approach developed by West et al. (2014) and implemented globally in Le Page et al. (2016). SSP4 is a world of inequality, both within and across regions. High-income regions continue to prosper, with increased

demand for energy and food. Technological progress, including agricultural productivity, is high. Low-income regions, however, stagnate; increases in total consumption are due to increased population and not increased wealth. Agricultural productivity growth is low. Environmental policies, including reduced deforestation, reforestation, and afforestation programs, are present in high- and medium income countries only. The SSP4-60 scenario includes modest expansion of global crop and pasture land, with increases of 14% and 9% from 2010 to 2100, respectively.

The modest climate policy encourages afforestation in the high- and medium-income regions where environmental policies are strong, resulting in a global increase in forest cover of 3% between 2010 and 2100.



### 2.8.6 SSP4-3.4 GCAM

The SSP4-3.4 scenario starts from the same baseline as the SSP4-60, but includes a more stringent mitigation policy limiting radiative forcing to 3.4 Wm⁻² in 2100. SSP4-3.4 was also simulated with GCAM (described above). Limiting 2100 radiative forcing to 3.4 W/m2 requires a much larger carbon price, exceeding $1000/tCO2 (2005 US$) in 2100, than the SSP4-60. This increased carbon price has substantial effects on energy and land use. In particular, ~1200 million ha of land is allocated to the production of bioenergy, resulting in a large increase in total cropland area (80% increase between 2010 and 2100). Forest cover increases in the high and medium-income
regions as the result of afforestation policies but decreases in the low-income regions as the result of agricultural land expansion. The net effect is that global forest cover increases through mid-century before returning to 2010 levels at the end of the century.

### 2.8.7 SSP5-3.4OS REMIND-MAGPIE

The SSP5-3.4OS scenario starts from the baseline SSP5-RCP8.5, but includes mitigation policy limiting radiative
forcing to 3.4 Wm⁻² in 2100.  SSP5 RCP3.4OS was also simulated with REMIND-MAgPIE (described above) (Kriegler et al., 2017). This scenario is supposed to follow SSP5-8.5, an unmitigated baseline scenario, through 2040, but includes after 2040 strong mitigation action to rapidly reduce CO2 emissions to zero around 2070 and to net negative levels thereafter. In consequence, the SSP5-RCP3.4OS pathway shows even stronger cropland expansion compared to the SSP5-RCP8.5 scenario, mainly due large-scale deployment of 2nd generation bioenergy
crops after 2040. Globally, cropland in the SSP5-RCP3.4OS pathway increases by about 800 Mha (50%) between 2010 and 2100, mainly at the cost of pasture area.

### 2.8.8 SSP1-1.9 IMAGE

The SSP1-1.9 parallels SSP1-2.6 in all aspects, but reaches a lower radiative forcing target, namely 1.9 instead of 2.6 W m⁻². As SSP1-2.6, also SSP1-1.9 is derived from the IMAGE 3.0 integrated assessment model (Stehfest et al.,
2014). IMAGE is a model framework describing the future agriculture system and energy system, as well the changes in future land cover, the carbon and hydrological cycle and climate change, as described above. The SSP1-1.9 is based on the SSP1 baseline scenario. As also described above, SSP1 projects a future under a green growth paradigm, with moderate population growth, and fast economic growth and technological improvements (van Vuuren et al, 2017). In terms of land use, SSP1 describes a world that is environmentally aware, and aims at limiting biodiversity loss and
environmental impacts of food consumption. Mitigation policy is added to the SSP1 baseline scenario to limit warming to 1.9 W m⁻² (Rogelj et al., 2018; Doelman et al., 2018). As for SSP1-2.6, important policies from the land-use perspective are increased bio-energy use in combination with carbon capture and storage, avoided deforestation policy to reduce deforestation, and restoration of degraded forests (Doelman et al., 2018).





**2.9 Harmonization of Inputs**

Harmonization of inputs involved minimizing the difference between the end of the historical reconstruction and the beginning of future projections, and preserving as much information on the future from IAMs as possible. Five different IAMs provide future land-use, wood harvest, and management data using a variety of variables and units and at different spatial and temporal resolutions (Table 2). Prior to harmonization, inconsistencies in definitions,

resolutions, and other factors resulted in significant discrepancies. The spread of global cropland values from the IAMs in 2010 was 5% of the historical reconstruction values in that year, and the spread of global pasture values from the IAMs in 2010 was 23% of the historical values. Gridded values had even larger discrepancies, differing by as much as 100% from the historical values. After harmonization, these inconsistencies were eliminated by design of the harmonization methodology. Since some IAMs didn't simulate built-up area or urban spread, and for consistency

of urban-land definitions across all scenarios, the IMAGE model provided land-use inputs for built-up area in all scenarios (Doelman et al., 2018). Also, since the REMIND-MAGPIE model did not compute wood harvest amounts, these were provided for the SSP5-8.5 and SSP5-3.4OS scenarios by the GCAM model.

The first step in harmonizing inputs was to convert the IAM data into a standardized format for comparison with the historical product. Future land-use data were aggregated into the fractions of each grid-cell occupied by total

cropland, total grazing land, urban land, and natural vegetation (the sum of primary and secondary forest and non-forest) annually at 0.25°×0.25°resolution. Future data on irrigation and flooded areas were standardized into national totals. Future wood harvest data were standardized into a total national wood harvest demand in Mg C y$^{-1}$, as well as the fuelwood component of that national wood harvest, either by aggregating gridded wood harvest data into national totals, or by disaggregating regional wood harvest data using the ratio of national to regional wood harvest

from the end-of-historical period (i.e. 2015). Wood harvest data that were provided in volume units (m$^3$) were converted to biomass (Mg C) using a conversion factor of 0.2688 Mg C m$^{-3}$. A 30% slash fraction was added to wood harvest scenarios that did not already include slash. Future fertilizer rates were standardized into national fertilizer application rates in kg N ha$^{-1}$ y$^{-1}$ per crop functional type. For future scenarios with only regional data, all countries within a region were assigned the same regional rates. When gridded future fertilizer application rates

were available these were also used in LUH2 and were standardized into annual rates per crop type (kg N ha$^{-1}$ y$^{-1}$) at 0.25° × 0.25° resolution. For SSP4-3.4 and SSP4-6.0 (both from GCAM), the fertilizer rates for the GCAM crop types *misccrop* and *palmfruit* were used as estimates of fertilizer rates for C3 perennials, *sugarcrop* and *biomass* rates were used as estimates for C4 perennial rates, *oilcrop* and *misccrop* rates were used for C3 nitrogen fixing crops, *rice* and *wheat* were used for C3 annuals, and *corn* was used for C4 annuals.

Although the IAM land-use data were generally in good agreement with the end-of-historical period values at the global scale, there were still significant differences both globally and spatially (Fig. 4). To address this issue, we applied IAM-based annual changes in land use sequentially to the spatial pattern of land use at the end of the historical reconstruction. Annual future changes in cropland, grazing land, and urban land were computed and aggregated to 2°x2°. These changes were then applied to the 2°-aggregated cropland, grazing land, and urban land,

from the previous time-step, starting with the end-of-historical period (i.e. 2015). When it was not possible to apply the annual change within a 2° grid-cell, due to lack of available land to expand into, or lack of cropland, grazing, or urban land to abandon, the unmet changes were applied in neighboring 2° grid-cells, starting with immediate neighbors and then radiating outward. The harmonized grids of cropland, grazing land, and urban land were then disaggregated into 0.25°×0.25° grids according to the following method: when disaggregating decreases, the

percentage change in each land-use state was computed and then applied to all underlying 0.25° land-use fractions; for increases in cropland, grazing, or urban land, the needed change was applied across all underlying 0.25° grid-cells and was weighted by available land in each grid-cell. Figure 5 shows how well the IAM 2015-2100 changes in cropland and pasture fractions are retained in the harmonized data, which increases markedly with decreased spatial resolution. For wood harvest, analogous methods were applied.

After the harmonization of total cropland, grazing land, and urban land, cropland and grazing areas were further disaggregated into underlying sub-types. Assignment of future crop functional types were based on fixed contemporary Monfreda/FAO proportions, and adjusted to match IAM specific information as needed. For grazing land, a pasture/rangeland mask was generated for 2015 (and held constant for all years) to sub-divide future total grazing land into the two grazing sub-types. For new grid cells projected to be converted to grazing land in the

future, national ratios were used.

Next, management data were harmonized by applying analogous algorithms to sequentially apply projected changes in managed area and rates to the pattern at the end of the historical reconstruction. Annual change in national irrigated areas were computed and then applied to the previous years gridded irrigation fractions for all crop types, first increasing irrigated area on grid-cells with existing irrigation, and then adding any additional needed irrigated

area equally to all non-irrigated cropland grid-cells within each country. Annual national percentage change in flooded area was computed and this percentage change was applied to all grid-cells that have a non-zero flooded fraction in the previous time-step. Any resulting fractions that are greater than 1 are reset to 1. Finally, annual national percentage changes in fertilizer rates per crop type are computed. These national percentage changes are applied to the previous years gridded fertilizer rates for all grid-cells within each country, with the resulting rates

held between 0 and 500 kg N ha$^{-1}$ yr$^{-1}$. In an effort to ensure that the final (year 2100) gridded fertilizer rates closely approximate the future IAM fertilizer rates, there are a few exceptions to this method, which aim to keep the LUH2 rates from remaining too low, or becoming too large, when compared to the IAM gridded rates. First, for grid-cells with fertilizer rates below 1 kg N ha$^{-1}$yr$^{-1}$ on the previous time-step, and with an increasing national percentage change in fertilizer rates, the actual gridded IAM fertilizer rates for the next time step are used instead of the

computed LUH2 rates. Also, if gridded fertilizer rates increase between time-steps and are above the gridded IAM fertilizer rates, the gridded fertilizer rates for the next time-step are held constant at the current LUH2 gridded rates. Finally, if the gridded LUH2 fertilizer rates are less than 80% of the IAM gridded fertilizer rates, and the national percentage change in fertilizer rates is positive, a small additional increase (1% of the total current difference between IAM gridded rates and LUH2 gridded rates) is added to the LUH2 fertilizer rates.





### 2.10 Additional Major Factors

#### 2.10.1 Inclusiveness of Wood Harvest

Since it is not always known whether or not the wood cut on land cleared for agriculture is counted in national wood harvest statistics, assumptions are made in LUH2 about the amount of biomass from land clearing that is included towards meeting national wood harvest demands. The need to use wood from cleared land for fuel or wood products was probably higher in the past than it is now. To that end, we assumed all wood on land cleared for agriculture prior to 1850 was counted towards meeting the national wood harvest estimates and additional wood harvest was only conducted when the land cleared for agriculture did not provide enough wood to meet the estimates. We also assumed that after 1920 none of the wood from cleared land was counted toward meeting national wood harvest numbers and wood harvest demand was met only through explicit wood harvesting activities. Between 1850 and 1920 a fraction of the wood from cleared land was used to meet wood harvest demands, starting from 100% of wood from cleared lands in 1850 and decreasing linearly to 0% in 1920. If this fraction of wood from cleared lands was not enough to meet national wood harvest demands, additional explicit wood harvest was conducted to meet national totals.

#### 2.10.2 Priority of Land Conversion

When converting natural land to agriculture, or using it for wood harvest, a decision must be made about whether to prioritize the use of primary or secondary land. The cumulative effect of these decisions has a large impact on the resulting secondary land area, age, and biomass in each grid-cell, and in aggregate at the regional and global scale. Although the decision of which natural vegetation type to prioritize is undoubtedly variable in space and time, for the sake of simplicity we have chosen a single priority rule for each land-use transition type, as follows. For urban expansion, secondary was prioritized. After all secondary land is used, further urban land-use demand (if any) was met on primary land. For expansion of cropland and grazing land, both primary and secondary land were used in relative proportion to their availability in each grid-cell. For example, if primary land and secondary land occupied 10% and 90% of natural vegetation in a grid-cell, respectively, then 10% of the converted natural vegetation would be taken from primary land, and 90% of the converted natural vegetation land would be taken from secondary land. For shifting cultivation, secondary land was prioritized unless the secondary land area was less than 10 times the cropland area in a grid-cell, in which case primary land was prioritized. For wood harvesting, the priority was to take wood from both primary and secondary land in relative proportion to the amount of available biomass in each land type.





### 2.11 Methodology for Calculating Land Use Transitions

**2.11.1 Determining agriculture land use transitions**

Following Hurtt et al. (2011), a book-keeping approach was used to calculate annual land-use transition rates between five aggregate land-use types—cropland, grazing land, urban, primary and secondary. To determine these, the annual change in urban area in each grid cell was first computed from either the HYDE data (for the historical period) or IAM data (for the future period) and applied proportionally to the cropland, grazing land, and secondary

land-use categories within the grid cell. If there was not enough land available between cropland, grazing land and secondary land for a given urban land-use increase, the remaining area needed was taken from the primary land within the grid cell. Next, minimum transition rates were calculated between the remaining three land-use types (cropland, grazing land, and other; where other was defined as the sum of primary and secondary), based on the gridded annual input data on land-use patterns from HYDE or the IAMs (adjusted for the transitions into and out of those types associated with urban land- use change computed on the previous step). With only three land-use types,

unique minimum transitions (i.e. solutions to Eq. 1) could be easily determined. Additional transitions associated with shifting cultivation and wood harvest were then determined. In cases of shifting cultivation, land-use transitions from cropland to other, and other to cropland, were both increased by the abandonment rate of agricultural land. Transitions from other were then partitioned into transitions from primary and secondary based on availability and

the previously described shifting cultivation algorithm. All transitions from cropland or grazing land to other were defined as transitions to secondary. The amount of wood cut in converting land to agriculture was determined by overlaying these transitions with estimates of biomass density.



After computing transitions between the five aggregate land-use types, the transitions to/from both primary and secondary were further sub-divided into transitions to/from primary forest, primary non-forest, secondary forest, and secondary non-forest, based on the underlying map of potential forest (grid-cells with potential biomass density greater than 2 kg C m$^{-2}$ were designated as potentially forested). In addition, the transitions to/from grazing land were subdivided into transitions to/from managed pasture and rangeland, based on the annual gridded input data from HYDE. The HYDE maps of managed pasture and rangeland for the year 2015 were also used to sub-divide

grazing land into the underlying grazing sub-types for all years in the future period (2015-2100). Transitions to/from total cropland in each grid-cell were further sub-divided into transitions to/from each of the five crop functional types (CFTs) using the data and methodology described in the section on "Historical Maps of Crop Types and Crop Rotations".

**2.11.2 Determining area cleared by wood harvest**

Since the spatial patterns of wood harvest within each country are not generally known (especially for years outside the period of satellite observations), several assumptions were used to spatially allocate the reconstructed national annual wood harvest demands to individual grid-cells within each country, and to convert the biomass harvested to an area cleared per grid-cell. As a first step, within each country and at each time-step, a fraction of the biomass

cleared from agricultural land expansion is subtracted from the national wood harvest demand, as described in the preceding section on the inclusiveness of wood harvest data. After wood from agricultural clearing has been subtracted, the remaining national wood demand is then explicitly harvested, first from grid-cells with available primary forest and/or mature secondary forest, then from grid-cells with young secondary forest, and finally from non-forested land (both primary and secondary). Mature secondary forests are defined using an average probability

of harvest vs. biomass function parameterized from detailed age-specific harvesting algorithms previously developed and applied in the U.S. (Hurtt et al., 2002; Hurtt et al. 2006). Note that since the natural vegetation definitions are based on a *mean* biomass density, wood harvesting from non-forested land can imply either harvesting vegetation, such as shrubland, that is tree-based albeit with a mean biomass density below that of a forest, or harvesting isolated trees within other low-biomass-density vegetation such as grasslands.

Within the group of grid-cells containing primary forest and/or mature secondary forest in each country, the first cells to be harvested are all those with a "significant human presence" (SHP), followed by all neighboring cells, radiating outwards, taking only the fraction of biomass needed until the demand has been satisfied or the available biomass exhausted. The use of proximity to a SHP in this algorithm is based on the assumption that proximity to a SHP implies proximity to transportation infrastructure (accessibility) or local markets. Prior to the year 1900, grid-

cells with a SHP are defined as those grid-cells having cropland, managed pasture, secondary land, or urban land area. Grid-cells that have Landsat-observed forest loss of at least 10% of the cell's land area during the period 2000-2012 are gradually included in the definition of SHP between the years 1900 and 2000, until both the land-use-based



and Landsat-based definitions of SHP are given equal weighting between 2000 and 2015. The contribution of Landsat-based forest loss to SHP then decreases again between 2015 and 2100.

When harvesting wood from a grid-cell chosen using these methods, if only a fraction of the biomass in a grid-cell is needed, wood is harvested from both primary forest and secondary mature forest (or from primary non-forest and secondary non-forest) in proportion to their available biomass. Wood harvested from primary land provides an area-based transition "primary to secondary", whereas wood harvested from secondary land provides an age- (and biomass-) resetting transition "secondary to secondary". To calculate these transitions in area units, the wood harvest
biomass was converted using the carbon density of land affected (Hurtt et al. 2006).

In addition to its use in the definition of SHP, the Landsat forest loss data is also used in two additional ways to further constrain the spatial pattern of wood harvesting. First, primary forest and mature secondary forest land that will experience a Landsat-observed forest loss during the period 2000-2012 is protected from wood harvest between the years 1950 and 2000 so that it is available for harvesting during the period 2000-2012. Second, during the years
2000-2012, the Landsat forest loss data is used in LUH2 to constrain the spatial pattern of where wood harvest does, or does not, occur, by checking whether the annualized gridded forest loss from the Landsat data has already been met within LUH2 yet. Inclusion of Landsat-based forest loss data in the LUH2 algorithm generates a significant improvement in the match between satellite observations of forest loss and the LUH2 representation of forest loss between the years 2000-2012 (Fig. 6).

For European countries that are unable to meet their national wood harvest demand with the available biomass, the unmet wood harvest from each country is reassigned to other European countries (including the former USSR), proportional to available biomass, and the spatial pattern of this additional wood harvest is then allocated using the same rules as outlined above. This is done to model the known trade in wood that was occurring between European countries, even in the early years of our historical simulation (Kaplan et al., 2017).

**2.12 Added Tree Cover**

While it is primarily a land use dataset, LUH2 does also provide a simple estimate of forest cover change. For IAM future scenarios with positive forest cover gain (SSP1-2.6, SSP2-4.5, SSP1-1.9), an algorithm was developed to match the spatial pattern of forest gain from IAMs, preserve existing harmonized land-use transitions, and that could be implemented relatively easily in ESMs. For each scenario, a supplementary file was created with a data variable
called 'added_tree_cover'. The variable specifies the added tree cover that needs to be planted in each grid cell each year to better represent the corresponding IAM Added Tree Cover estimates. For the other IAM scenarios that are not affected by this issue, added_tree_cover values are set to zero. To produce these datasets, the spatial pattern of differences in forest cover between LUH2 and each corresponding IAM were computed annually for 2015-2100. For each year, each grid cell, if the difference could be met on LUH2 classified non-forest land, that difference was
noted as 'added_tree_cover' in the new file. If the gain could not be met on the non-forest area, the change was applied on nearby cells up to 4 grid cells away.



### 2.13 Extensions 2100-2300

In addition to the eight future scenarios for the period 2015-2100, the LUH2 dataset also includes extensions for the years 2100-2300 for three of the harmonized future land-use forcing datasets for use in long-term climate stabilization experiments. By design, in these extensions, all land-use states and management variables are held constant at year 2100 values for the years 2100-2300. As a result, almost all transitions between land-use states are set to zero, with the exception of crop rotations and shifting cultivation, which continue at their year 2100 rates, and

wood harvest, which uses year 2099 national wood harvest demands for all years from 2100 to 2299. These extensions to future scenarios are available for SSP1-2.6, SSP5-3.4OS, and SSP5-8.5.

## 3    Results

### 3.1 Aggregate Results

The annual, gridded land-use states are aggregated to annual global values by multiplying the grid-cell land-use

fractions by the grid-cell area and summing over all grid-cells (Fig. 7). These 12 states can be further aggregated into the 5 broader land-use categories of total cropland (the sum of all 5 crop types), total grazing land (the sum of managed pasture and rangeland), primary land (the sum of primary forest and primary non-forest), secondary land (the sum of secondary forest and secondary non-forest), and urban land. Historically, the area of cropland increased at an accelerating rate from $1.7 \times 10^6$ km$^2$ in 850, to $4.3 \times 10^6$ km$^2$ in 1800, and $15.9 \times 10^6$ km$^2$ by 2015 (Fig. 7).

Grazing lands increased more rapidly, from $3.3 \times 10^6$ km$^2$ in 850, to $9.2 \times 10^6$ km$^2$ in 1800, and to $32.8 \times 10^6$ km$^2$ by 2015. Urban increased from 0 in 850 to $0.6 \times 10^6$ km$^2$ by 2015. During the historical period (850-2015 CE), primary land area decreased from $125 \times 10^6$ km$^2$ to $50.1 \times 10^6$ km$^2$ (of which 44% is forested), while secondary land increased from 0 to $30.4 \times 10^6$ km$^2$ (of which approximately 49% is forested); note that by definition LUH2 initializes secondary land area to zero in 850 CE. The new land-use history reconstruction derived here generally compared

favorably to prior reconstructions (Hurtt et al, 2006; Hurtt et al., 2011) and other references across a range of important diagnostics (Table 3), albeit at higher spatial resolution and with more process detail.

For the future, all eight scenarios projected increases in global cropland area, while six projected grazing land decreases (SSP4 RCP6.0 from GCAM, and SSP3 RCP7.0 from AIM projected grazing land increases). However, six out of eight scenarios projected large increases in wood harvesting, which contributed to large increases in

secondary area and corresponding reductions in primary area by 2100. In 2100 global cropland ranged from $17.8 \times 10^6$ km$^2$ (SSP1 RCP2.6 from IMAGE) to $29.1 \times 10^6$ km$^2$ (SSP4 RCP3.4 from GCAM). As shown in Table 4 and Figure 15 (panel a), for 6 out of 8 scenarios the dominant crop functional type in 2100 was C3 annuals, with C4 perennials (for biofuels) the dominant crop functional type in 2100 for the remaining two scenarios (SSP4 RCP3.4 from GCAM and SSP5 RCP3.4OS from REMIND-MAGPIE). Global grazing land in 2100 ranged from $25.4 \times 10^6$

km$^2$ to $35.5 \times 10^6$ km$^2$, with the majority of that coming from rangeland (Table 4). Secondary land in 2100 ranged





from $36.5 \times 10^6$ km$^2$ to $44.5 \times 10^6$ km$^2$ (Table 4). In all cases, approximately half of all secondary land was forested, and the estimated mean age of secondary forest ranged from 58 yr to 74 yr. Added tree cover data layers, were computed to match the forest tree cover gains of the SSP1-2.6, SSP2-4.5, and SSP1-1.9 scenarios and were able to capture >80% of the global afforestation signal in the IAM scenarios. Extensions to year 2300 were computed for the SSP1-2.6, SSP5-3.4OS, and SSP5-8.5 scenarios, and by design did not change the gridded or global cropland, grazing land, or urban land areas. However, due to wood harvesting and shifting cultivation continuing at their end-of-century rates, the area of secondary vegetation continued to grow, and the area of primary vegetation continued to decline in these extensions. By 2300 the global secondary vegetation area in these extension scenarios ranged between $46.3 \times 10^6$ km$^2$ and $51.2 \times 10^6$ km$^2$, while the global primary vegetation area ranged between $28.6 \times 10^6$ km$^2$ and $33.0 \times 10^6$ km$^2$.

Gross transitions (the sum of the absolute value of all land-use transitions) are a measure of all land-use change activity. In general, the annual gross transitions tend to increase through time, beginning at $2 \times 10^5$ km$^2$ in 850 and increasing to $1.86 \times 10^6$ km$^2$ in 2000 (Table 3). The differences between the historical period lower, baseline, and upper scenarios in LUH2 prior to 1920 are primarily due to the differences in rates of wood harvest between those three scenarios. After 1920 the three LUH2 historical scenarios share the same wood harvest reconstruction and their associated gross transitions are very similar. In the future scenarios, gross transitions mostly increased and by 2100 ranged from $2.0 \times 10^6$ km$^2$ to $4.8 \times 10^6$ km$^2$ (Table 5).

Net transitions measure only the net changes into land use (excluding wood harvest on secondary forests, shifting cultivation, and other agricultural land abandonment that is offset by land conversions to agriculture). Net transitions increase from $2 \times 10^4$ km$^2$ in 850 to $2.3 \times 10^5$ km$^2$ in 2000 (Table 3). The net transitions across all three historical LUH2 scenarios (lower, baseline, and upper) are all very similar at most time points. The LUH2 historical scenario shows a significant reduction in transitions to pasture around 1950-1960, with implications for carbon investigated separately (Ma et al., In Review). In the future, net transitions range from $-1.1 \times 10^5$ km$^2$ to $1.6 \times 10^5$ km$^2$ in 2100 (Table 5).

To visualize the magnitudes of transitions between variables, we present badge plots indicating the average net transitions occurring annually from 850-1849, 1850-2015, 850 – 2015, as well as 2015-2099 for all future scenarios amongst all the major land-use categories (Fig. 8). Each arc in a badge plot represents the average annual area transitioning from one land-use to another. The color of the arc represents the land-use category from which transition occurs to a different category. For example, in Figure 8 the arc in light green color represents the transition from cropland to other categories. Transitions involving croplands and secondary forest lands dominate land-use transitions in all three scenarios. The dominant land-use transition is secondary forest lands to croplands and it ranges from nearly $6 \times 10^4$ km$^2$ y$^{-1}$ in the low historical scenario to $8 \times 10^4$ km$^2$ y$^{-1}$ in the baseline scenario and $1 \times 10^5$ km$^2$ y$^{-1}$ in the upper scenario when averaged from 850 – 2015. Cropland abandonment activities are also significant with nearly $1 \times 10^5$ km$^2$, $1.4 \times 10^5$ km$^2$ and $1.7 \times 10^5$ km$^2$ of croplands transitioning annually to secondary lands (both forested and non-forested) in the low, baseline and high LUH2 historical scenarios respectively





(averaged over the entire historical period). On an annual basis, the transitions to and from croplands and secondary lands are generally the same in all three LUH2 historical scenarios.

LUH2 historical results were compared to multiple diagnostics (Table 3). Almost all metrics are within, or very close to published reference ranges. These metrics show that 65% of the secondary land increase between 1700 and 2000 is forested, and 93% of U.S. forests in the year 2000 are on secondary land. Global natural vegetation in biodiversity hotspots in the year 2005 is estimated as 1.6% of the land surface (compared with the reference value of 2.3%). The mean age of secondary land can be calculated for each grid cell and aggregated to a global mean age. For the first several hundred years of the simulation the global mean secondary age grew with time, due to primary land being used for land conversion and wood harvesting more often than secondary land (which was initialized to have zero area). Around 1700-1800, existing secondary land was used more often for new land conversions and wood harvesting and the global mean secondary age started to decrease with time. The median age of secondary forests in the year 2005 is 42 years, and is 43 years in the year 2015 (compared with the reference range of 30-40 years). The upper scenario had the highest secondary mean age, because it had a larger secondary land area, which allows that secondary land to be used less frequently for wood harvesting and land conversions. Conversely, the lower scenario had a lower secondary mean age than the baseline scenario. The overall land area impacted by human land use in the year 2000 is 59% of the land surface. The global area of secondary land increase between 1700 and 2000 is estimated as 13.2 $\times 10^6$ km$^2$ with 10.4 $\times 10^6$ km$^2$ of that area forested and 2.8 $\times 10^6$ km$^2$ non-forested.

Cumulative clearing for cropland and pasture between the years 1500 and 1990 resulted in 251 Pg C of wood removed (compared with a reference range of 121.9 to 356.3 Pg C). Total wood harvest over this period was 170 Pg C, of which 132 Pg C was from direct wood harvest and 38 Pg C was included from agricultural clearing. In the year 2000, an estimated 0.32 $\times 10^6$ km$^2$ of agricultural land was involved in shifting cultivation (compared with a reference value of 0.3 $\times 10^6$ km$^2$. Potential forest area 47 $\times 10^6$ km$^2$, compared to a reference value of 52 $\times 10^6$ km$^2$, and in the year 2015 global forest area was estimated at 37 $\times 10^6$ km$^2$, compared with a reference range of 32-41 $\times 10^6$ km$^2$. In the year 2000 global wood harvest was 1.29 Pg C, of which 0.71 Pg C was for fuelwood. Global synthetic fertilizer usage in the year 2012 was 106.6 Tg N yr$^{-1}$ (compared with a reference value of 100 Pg C), and the global area of irrigated cropland in 2003 was 2.51 $\times 10^6$ km$^2$ (compared with a reference value of 2.77 $\times 10^6$ km$^2$). In 2004, the area of cropland (primarily corn) used for biofuels was 0.03 $\times 10^6$ km$^2$ compared to the reference value of 0.033 $\times 10^6$ km$^2$. Total potential plant biomass on all lands was 718 Pg C (compared with a reference range between 557 and 923 Pg C), while total plant biomass in 2005 was 434 Pg C (compared with a reference value of 393 Pg C). Plant above-ground biomass on pantropical forested lands between years 2007-2008 was 184 PgC (compared with a reference range between 188 and 229 PgC), and total plant biomass on forested lands in 2005 was 395 (compared with a reference value of 363 Pg C). In addition, the cumulative loss of above-ground biomass resulting from land-use transitions (i.e., the sum of all losses) is an important metric of the gross effects of land use on the terrestrial carbon cycle and rose from 0 Pg C in 850 to 5.6$\times 10^4$ Pg C in 2015. Similarly, the cumulative net loss in above- ground biomass is the difference between the estimated above-ground biomass including land use, and the estimated biomass of potential vegetation, and includes both the losses of above-ground biomass due to



land-use and the gains due to regrowth. During the historical period the global cumulative net loss of above ground biomass carbon increases monotonically from nearly zero in 850AD to around 310 Pg C in 2015. The lower, baseline, and upper historical scenarios all give similar global estimates of this metric; the upper scenario gives the
790 highest estimates, which is presumably due to the high historical wood harvest in this scenario.

In the future scenarios secondary land increases between 6.0% and 13.27% across the years 2015 to 2100, with between 48.91% and 72.81% of that increase being on potentially forested land (Table 5). The median age of secondary forest in the year 2100 ranges between 58. and 74 years. The global area covered by natural vegetation in the biodiversity hotspots ranges between 0.57% and 1.08% of the land surface. Wood clearing for cropland and
795 pastures across the years 2015 to 2100 removes between 44 and 88 Pg C of above ground biomass, whereas direct wood harvest removes between 93 and 148 Pg C of above ground biomass. Global wood harvest in the year 2100 ranged between 0.9 and 1.87 Pg C, of which the fuelwood component was between 0.15 and 0.88 Pg C. Total forest area change between 2015 and 2100 ranged from a decrease of $5.1 \times 10^6 \, \text{km}^2$ to an increase of $3.42 \times 10^6 \, \text{km}^2$, resulting in a global forest area in 2100 of between 32.1 and $38.1 \times 10^6 \, \text{km}^2$. Global fertilizer use in the year 2100
ranged between 110 Tg N yr$^{-1}$ and 240 Tg N yr$^{-1}$, while the global irrigated area in 2100 ranged between 2.6 and 4.1 $\times 10^6 \, \text{km}^2$. Land flooded for rice in 2100 ranged from 0.23 to $0.96 \times 10^6 \, \text{km}^2$, and cropland used for growing biofuels in 2100 ranged from 0 to $18 \times 10^6 \, \text{km}^2$. Total biomass of natural vegetation on forested lands in 2100 ranged between 290 and 391 Pg C, of which between 170 and 239 Pg C is above ground biomass on pantropical forested lands. In 2100, the global cumulative net loss of above ground biomass carbon ranges widely across
scenarios, from 320 Pg C to 385 Pg C.

### 3.2 Spatio-temporal Patterns of Land Use Transitions, Secondary Area, and Secondary Age

Regional results for the historical period, averaged for each century, are shown in Table 6. In each region or continent, secondary land, gross transitions, and net transitions all tended to increase with time. Secondary land, along with both gross and net transitions, was highest in Eurasia and Africa. Mean regional secondary land area was
810 $8.47 \times 10^6 \, \text{km}^2$ in Eurasia and $6.01 \times 10^6 \, \text{km}^2$ in Africa in the 1700s and increased to $12.43 \times 10^6 \, \text{km}^2$ and $6.82 \times 10^6 \, \text{km}^2$ in Eurasia and Africa respectively in the 1900s. Gross transitions peaked in Eurasia in the 1800s at $660 \times 10^6 \, \text{km}^2$ yr$^{-1}$, while net transitions peaked in Eurasia in the 1900s at $121 \times 10^6 \, \text{km}^2$ yr$^{-1}$. After 1700, secondary age tended to decrease with time for most regions, although it has held relatively constant over the last three centuries for both Africa and Oceania. The range of secondary mean age in the 1900s was between 52 years to 289 years. In 1850
there are large areas of cropland in the Eastern USA, Europe, India, and China, and large areas of primary land world-wide with the exception of Europe, Northern Africa and the Middle-East (Fig. 9). By 2015 cropland areas have expanded through-out Africa and the Americas as well, primary land is lost in large areas of the Eastern USA, Africa, Europe, India, and China, and mean secondary age is lower in most locations (Fig. 10).

Regional results are also averaged for the period 2000-2099 for each future scenario (Table 7). Across all scenarios,
there were only small differences in regional secondary areas (3.8-4.5$\times 10^6 \, \text{km}^2$ for North America, 2.0-3.0$\times 10^6 \, \text{km}^2$ for South America, 17-18$\times 10^6 \, \text{km}^2$ for Eurasia, 9.2-11$\times 10^6 \, \text{km}^2$ for Africa, and 0.7-0.87$\times 10^6 \, \text{km}^2$ for Oceania) with



SSP1-1.9 having the highest secondary area in each continent. Secondary land area was highest in Eurasia and Africa for all scenarios. Regional secondary age also did not vary significantly across scenarios; the SSP5-8.5 scenario had the highest secondary age for all regions except Oceania (67 years for North America, 49 years for South America, 209 years for Eurasia, 70 years for Africa, and 50 years for Oceania) and the SSP4-3.4 scenario had the lowest secondary age for most regions (60 years for North America, 45 years for South America, 197 years for Eurasia, 69 years for Africa, and 48 years for Oceania). Secondary age was highest in Eurasia for all scenarios. Gross transitions were highest in Eurasia in 7 out of 8 scenarios (with Africa the second highest), and highest in Africa in one scenario (with Eurasia the second highest). The highest overall rate of gross transitions was $1936\times10^6$ km$^2$ yr$^{-1}$ in Eurasia in the SSP5-3.4OS scenario, but comparable rates of gross transitions were also observed in Eurasia and/or Africa in the SSP4-3.4, SSP4-6.0, SSP3-7.0, and SSP5-8.5 scenarios. Net transitions were largest in Africa in all scenarios (between 34-143$\times10^6$ km$^2$ yr$^{-1}$) and lowest in Oceania in 7 out of 8 scenarios (and negative in 6 of those), with South America having the lowest net transitions in the remaining scenario. The SSP4-3.4, SSP4-6.0, and SSP3-7.0 scenarios had the highest rates of net transitions overall at 143$\times10^6$ km$^2$ yr$^{-1}$, 133$\times10^6$ km$^2$ yr$^{-1}$, and 133$\times10^6$ km$^2$ yr$^{-1}$ respectively.

Large-scale spatial patterns are similar across most scenarios in the year 2100 (Figs 11-14), with the trends of increased cropland area in South America, continued loss of primary land worldwide and particularly in Africa, and continued reduction of mean secondary age. Analogous mapped results for Tier 2 scenarios are provided in the Appendix.

### 3.3 Land-use Management

During the historical period, the use of synthetic nitrogen-based fertilizer on croplands was zero until the early 20$^{th}$ century. After 1950 fertilizer usage started increasing rapidly, and by 2015 global synthetic nitrogen fertilizer usage was 112 Tg N y$^{-1}$ (4153 Tg N cumulatively from 1915 to 2015; none prior to 1915), with the majority of this being applied in cropland-dominated locations including the North America, Europe, India, China, and South-East Asia. The eight harmonized future scenarios show a range of potential nitrogen futures; all except one scenario (the SSP5-8.5, which does increase but then falls again to close to current year values) project an increase in global nitrogen fertilizer usage. The range of harmonized global nitrogen fertilizer values in 2100 is between 110 Tg N y$^{-1}$ and 240 Tg N y$^{-1}$, with a total cumulative use of synthetic nitrogen fertilizer from 2015 to 2100 between 9840 Tg N and 14841 Tg N (Figure 15, panel b).

The global area of irrigated cropland increased steadily throughout the historical period and was around 2.7 million km$^2$ in 2015. The spatial patterns of this irrigated area show that the majority of global irrigation occurs in India and China, with other significant areas in the USA, Europe, Middle East, and South-East Asia. Six out of eight future scenarios project the global irrigated area to remain steady, or even decrease slightly, whereas two future scenarios (SSP3-7.0 from AIM and SSP5-8.5) show large increases in global irrigated area. The range of values across all future scenarios in 2100 is between 2.6 and 4.1 million km$^2$ (Figure 15, panel c).

The global use of croplands area for purpose-grown biofuels was very low prior to the year 2000 when a small amount of first generation biofuels production began (such as corn or sugarcane). In the future scenarios the fraction of cropland area grown for first generation biofuels was held constant, although underlying changes in cropland area resulted in some small increases or decreases in the total area of first generation biofuels. Second generation biofuel

area (such as miscanthus or switchgrass) expanded in each of the future scenarios, assumed to start from zero in 2015. Five of the eight scenarios (SSP1-1.9, SSP1-2.6, SSP4-3.4, SSP5-3.4OS, and SSP4-6.0) all showed significant increases in the area of second generation biofuels, while the remaining three scenarios has very little growth in this land management type. By the year 2100, global areas of biofuel crops ranged between 0 and 18 million km$^2$, and maps of the spatial distribution of total biofuels area (both first and second generation biofuels) show the dominant

locations to be the USA, Europe, China, non-Amazonian Brazil, and Argentina. Large expansion of secondary biofuels primarily occurred in South-East Asia, Eastern Europe and the former USSR, and the Middle East (Figure 15, panel d).

## 4   DISCUSSION

Land use is essential for meeting human needs for food, fuel, fiber, and shelter, but also affects the biogeochemistry,

biogeophysics, biodiversity, and climate of the Earth. Quantitatively understanding the effects of land-use activities on the Earth system requires that the best information on land use be incorporated into the best Earth system models. The strategy described here (LUH2) builds on the approach for harmonizing land-use patterns and transitions in CMIP5 (LUH1, Hurtt et al., 2011). This new version is completely updated with new inputs, and includes higher spatial resolution, increased detail, added management layers, new future scenarios, and a longer time domain - in

all more than a 50-fold increase in data from its predecessor. As such, it is designed to facilitate more complete and more consistent treatments of how land-use changes influence the Earth system past-present-future.

In comparison to LUH1 (Hurtt et al., 2011), the LUH2 land-use history is spatially, temporally and thematically richer than the previous reconstruction. While not strictly comparable for these reasons, comparing the two products to each other and across a wide range of diagnostics reveals some important quantitative similarities and differences.

Historically, the globally aggregated magnitudes of key land-use states (i.e., cropland, grazing area) and key land cover variables (forest area and biomass) are generally quite similar (<10% difference) over periods of overlap. Larger differences between these datasets are found in transitions, resulting secondary lands, and spatial patterns of land-use activities, where contemporary global gross transitions are reduced by ~35%, contemporary net transitions increased by ~35%, and estimated primary forest in biodiversity hotspots much closer to independent estimates

relative to LUH1 (Jantz et al., 2015). Considering the past, LUH2 begins in 850AD, 650 years earlier that LUH1. Considering the future, the set of 8 future scenarios included in LUH2 doubles that of LUH1, expanding the range of land-use forcing that can be considered and including additional cases.  Like LUH1, LUH2 also includes extensions to 2100-2300 with no net change in forcing over the interval. LUH2 also includes new Added-tree-cover data, to better reflect the changes in tree cover projected by IAMs in afforestation scenarios.





Since management was a new input in LUH2, we do not have comparable values from LUH1. However, the estimates from LUH2 for key management variables are close to empirical estimates and reflect major alterations of nutrient and water cycles, with implications for climate. For example, the ~100 Tg N y$^{-1}$ of industrial fertilizer use and irrigated area ~2.5 million km$^2$ by 2000 indicate major human impacts on the functioning of agro-ecosystems in addition to a general land-cover change metric. The inclusion of these activities here as part of the global

harmonized dataset is intended to facilitate their inclusion in future global climate assessments, harmonized, and together with other concurring land-use changes.

These LUH2 datasets are part of the official CMIP6 input4MIPs data collection, and are required forcing datasets for the DECK and historical climate simulations (Meehl et al., 2014; Eyring et al., 2016). The data are also required for several of the CMIP6-MIP experiments including ScenarioMIP (O'Neill et al., 2016), LUMIP (Lawrence et al.,

2016), PMIP (Junclaus et al., 2017) and others. ScenarioMIP defined the set of future scenarios for consideration and organized the official climate-model experiment to quantify the effects of future scenarios of anthropogenic forcing on climate. LUMIP organized the set of model experiments focused on quantifying the effect of land-use forcing per se on climate. PMIP is organized to study the historical climate. The central use of these data in the DECK and across a range of important MIPs enhances consistency across CMIP6.

These datasets have also been adopted as required forcing for a range of other international studies including: ISIMIP (Frieler et al. 2017), Global Carbon Project (LeQuéré et al., 2018; Friedlingstein et al., 2019), and IPBES (Kim et al. 2018). The LUH2 datasets are regularly employed by the TRENDY modeling group in the annual carbon budget estimates of the Global Carbon Project using a simple linear interpolation to update to year of current budget (LeQuéré et al., 2018; Friedlingstein et al., 2019). The Global Carbon Project also provides a comparison of land use

and land use change emissions with quasi-independent data from 'bookkeeping' models that use FAO statistics directly. The bookkeeping and process-based model estimates of emissions tend to show high agreement, although in the last 3 years have begun to diverge (Friedlingstein et al., 2019). The additional data layers provided by LUH2 over earlier land cover reconstructions, which include information on land management such a wood harvest, prove interesting as studies have shown that the net land use change flux may have been substantially underestimated in

earlier studies that excluded land management effects on the carbon cycle (e.g., Arneth et al., 2017). This standardization of land-use forcing across the breadth of CMIP6 studies, and other international assessments has the promise to facilitate maximum consistency in the treatment of land use across the range of interdisciplinary foci and spatial/temporal domains of studies.

The LUH2 dataset was developed to provide globally consistent and coherent gridded land use for more than a millennium, spanning the past and future, as a necessary input for earth system model simulations for CMIP6. The requirement of global consistency through time means that it did not always incorporate all of the best local, regional, or national historical data available. For this reason, it may not necessarily be the optimal dataset for a local or regional analysis of land use impacts on biogeochemistry or biodiversity.

28





Looking ahead, ongoing CMIP6 and several other international activities will be engaged in using LUH2 data as input to studies of global climate, carbon, biodiversity and other assessments. These data products are intended to meet current needs of models, and also provide new variables that most models do not yet include but that may be important. Examples of these features include transitions, introduced in LUH1 and now a growing feature of many models, and now management variables.   Model development will need to continue to advance to utilize these

features. Meanwhile, advances need to proceed for the next generation of land-use harmonization. which should build on these advances and include additional data constraints, more process detail, and a focus on reducing uncertainty of the most sensitive features. This should be part of larger effort to develop a robust process to provide the best forcing data sets for future global assessments.

**Code Availability**

The source code used to produce the LUH2 historical datasets, along with the sources and citations of necessary inputs, are archived at https://doi.org/10.5281/zenodo.3533792. The source code used to produce the LUH2 future scenarios will also be made available and archived at zenodo.org.

**Data Availability**

The data produced in this study are archived and publicly available at the U.S. Department of Energy input4MIPS

site https://esgf-node.llnl.gov/search/input4mips/. The data are available in multiple files and fine-grain DOIs, and can be accessed and referenced using the following coarse grain citations, one historical (Hurtt et al., 2017) and one future (Hurtt et al., 2019).

**Author Contributions**

GH is the lead author and co-developed the method and conducted analyses with LC, RS, and SF. KKG, AH, JJ, JK, OM, JP, XZ provided historical input. BB, KC, JD, SF, TH, PH, FH, TK, AP, KR, ES, DV provided future scenario input.  JF, JK, DL, PL, LM, BP, ES, PT provided modeling input. FT provided input on FAO data. All authors contributed to writing the manuscript.

**Competing Interests**

The authors declare that they have no conflict of interest.






**Acknowledgements**

We gratefully acknowledge the support of the U.S. Department of Energy grant DESC0012972, and NASA grants

NNX13AK84A and 80NSSC17K0348. JP was supported by the German Research Foundation's Emmy Noether

Program (PO 1751/1-1). KKG was supported by Dutch NWO VENI grant no. 016.158.021. Part of the material in

the methods section is from Hurtt et al. 2011.





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



Table 1. Historical global population (millions) and land use estimates (million ha) from HYDE 3.2
(Klein Goldewijk et al., 2017).

|  | 800 CE | 1000 CE | 1500 CE | 1700 CE | 1850 CE | 1950 CE | 2015 CE |
|---|---|---|---|---|---|---|---|
| Population | 286 | 323 | 503 | 592 | 1271 | 2529 | 7301 |
| Cropland | 140 | 162 | 256 | 293 | 578 | 1223 | 1591 |
|     Rainfed area | 136 | 157 | 252 | 289 | 549 | 1118 | 1316 |
|     Irrigated area | 3.6 | 4.1 | 4.2 | 4.5 | 28 | 105 | 276 |
|     Rice area | 4.2 | 4.8 | 8.7 | 12.5 | 28 | 65 | 118 |
|         Paddy rice | 1.2 | 1.5 | 2.4 | 2.9 | 12 | 36 | 75 |
|         Rainfed rice | 2.9 | 3.3 | 6.3 | 9.6 | 16 | 29 | 43 |
| Grazing | 314 | 366 | 515 | 664 | 1192 | 2611 | 3241 |
|     Pasture | 31 | 55 | 105 | 145 | 253 | 535 | 787 |
|     Rangeland | 282 | 310 | 410 | 519 | 939 | 2076 | 2454 |
| % agric /total land area | 3.5% | 4.0% | 5.9% | 7.3% | 13.6% | 29.4% | 37.1% |





Table 2. Properties of SSPs used in this analysis. SSP-RCP refers to Shared Socioeconomic Pathway and
Representative Concentration Pathway, respectively and Tier refers to ScenarioMIP Tier (O'Neill et al., 2016).

| *SSP-RCP* | IAM | Tier | Crop | Grazing | Wood Harvest | Irrigation | Fertilizer |
|---|---|---|---|---|---|---|---|
| SSP5-8.5 | REMIND-MAGPIE | 1 | 0.5°x0.5° | 0.5°x0.5° | NA | 0.5°x0.5° | 0.5°x0.5° |
| SSP3-7 | AIM | 1 | 0.5°x0.5° | 0.5°x0.5° | 18 regions | 0.5°x0.5° | 18 regions |
| SSP2-4.5 | MESSAGE | 1 | 0.5°x0.5° | 30 regions | 0.5x0.5 | 30 regions | 30 regions |
| SSP1-2.6 | IMAGE | 1 | 0.5°x0.5° | 0.5°x0.5° | 26 regions | 0.5°x0.5° | 0.5°x0.5° |
| SSP4-6.0 | GCAM | 2 | 0.25°x0.25° | 33 regions | 33 regions | 33 regions | 33 regions |
| SSP4-3.4 | GCAM | 2 | 0.25°x0.25° | 33 regions | 33 regions | 33 regions | 33 regions |
| SSP5-3.4-OS | REMIND-MAGPIE | 2 | 0.5°x0.5° | 0.5°x0.5° | NA | 0.5°x0.5° | 0.5°x0.5° |
| SSP1-1.9 | IMAGE | 2 | 0.5°x0.5° | 0.5°x0.5° | 26 regions | 0.5°x0.5° | 0.5°x0.5° |






Table 3. Diagnostic table, historical data.

| Metric | Units | Time-period | Literature values | LUH2_v2h | LUH1 |
|---|---|---|---|---|---|
| **Transitions** | | | | | |
| Total gross transitions | $10^6$ km$^2$ yr$^{-1}$ | 2000 | | 1.86 | 2.9 |
| Total net transitions | $10^6$ km$^2$ yr$^{-1}$ | 2000 | | 0.23 | 0.17 |
| **Human land use impacts** | | | | | |
| Secondary land increase that is forested | % | 1700-2000 | | 64.5 | 57.6 |
| U.S. Forests that are secondary | % | 2000 | | 92.9 | 100.0 |
| Natural vegetation in biodiversity hotspots | % | 2005 | 2.3[1] | 1.6 | 4.6 |
| Median secondary forest mean age | yr | 2005 | | 42.2 | 27.6 |
| Median secondary forest mean age | yr | 2015 | 30–40[2] | 43.0 | |
| Land impacted by human land use | % | 2000 | | 58.7 | 54.0 |
| Secondary land area increase | $10^6$ km$^2$ | 1700-2000 | | 13 | 17 |
| Secondary land area increase (forest) | $10^6$ km$^2$ | 1700-2000 | | 10 | 10 |
| Secondary land area increase (non-forest) | $10^6$ km$^2$ | 1700-2000 | | 3 | 7 |
| **Wood harvest and agricultural clearing** | | | | | |
| | | 1500-1990 | 121.9– | 251 | 278 |
| Wood clearing for crop and pasture | Pg C | | 356.3[3] | | |
| Total wood harvest | Pg C | 1500-1990 | | 170 | |
| Direct wood harvest | Pg C | 1500-1990 | | 132 | 119 |
| Agricultural clearing for wood harvest | Pg C | 1500-1990 | | 38 | |
| **Shifting cultivation** | | | | | |
| Agricultural land for shifting cultivation | $10^6$ km$^2$ yr$^{-1}$ | 2000 | 0.3[4] | 0.3 | 0.6 |
| Agricultural land for shifting cultivation | $10^6$ km$^2$ yr$^{-1}$ | 1980 | 0.2-0.6[5] | 0.3 | 0.5 |
| **Forest loss and area** | | | | | |
| Potential forest area | $10^6$ km$^2$ | Potential | 48.7-55.3[6] | 47 | 51 |
| Forest area | $10^6$ km$^2$ | 2015 | 32.1-41.4[7] | 37 | |
| **Management** | | | | | |
| Fuelwood | Pg C | 2000 | *FAO value?* | 0.7 | |
| Wood-harvest | Pg C | 2000 | *FAO value?* | 1.3 | |
| Fertilizer use | Tg N yr$^{-1}$ | 2012 | 100[8] | 107 | |
| Irrigated area | $10^6$ km$^2$ | 2003 | 2.77[9] | 2.5 | |
| Biofuel area (corn, USA) | $10^6$ km$^2$ | 2004 | 0.033[10] | 0.03 | |
| **Biomass** | | | | | |
| Plant total biomass on all lands | Pg C | Potential | 557.4-923[11] | 718 | 731 |
| | | 2007-2008 | 187.5- | 184 | 177 |
| Plant AGB on pantropical forest lands | Pg C | | 228.7[12] | | |
| Plant total biomass on forest lands | Pg C | 2005 | 362.6[13] | 395 | 404 |
| Plant total biomass on all lands | Pg C | 2005 | 393.4[13] | 434 | 440 |

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






Table 4. Harmonized Scenarios of Future Land-use: global land-use state areas in year 2100, across all future scenarios ($10^6$ km$^2$).

|  | SSP1-1.9 | SSP1-2.6 | SSP4-3.4 | SSP5-3.4OS | SSP2-4.5 | SSP4-6.0 | SSP3-7.0 | SSP5-8.5 |
|---|---|---|---|---|---|---|---|---|
| C3 annuals | 7.86 | 7.94 | 9.13 | 7.72 | 10.44 | 8.39 | 10.52 | 9.04 |
| C4 annuals | 2.67 | 2.59 | 3.56 | 2.95 | 4.03 | 3.50 | 5.18 | 4.20 |
| C3 perennials | 2.78 | 2.79 | 2.95 | 2.22 | 2.02 | 1.82 | 2.17 | 1.59 |
| C4 perennials | 2.87 | 2.42 | 11.23 | 9.04 | 0.34 | 2.55 | 0.35 | 0.33 |
| C3 N-fixers | 2.11 | 2.11 | 2.27 | 2.11 | 3.03 | 2.38 | 3.34 | 2.77 |
| Managed pasture | 3.81 | 4.35 | 9.04 | 4.13 | 6.23 | 9.74 | 8.95 | 7.11 |
| Rangeland | 21.59 | 22.08 | 22.17 | 21.26 | 22.14 | 25.80 | 25.49 | 23.80 |
| Urban | 1.04 | 1.04 | 1.11 | 1.25 | 1.10 | 1.11 | 1.03 | 1.25 |
| Primary | 40.69 | 40.84 | 31.97 | 38.67 | 36.54 | 33.71 | 34.64 | 37.22 |
| Secondary | 44.51 | 43.77 | 36.50 | 40.59 | 44.05 | 40.95 | 38.27 | 42.62 |





Table 5. Diagnostic table, future land-use.

| Metric | Units | Time period | SSP1 RCP1.9 | SSP5 RCP3.4OS | SSP1 RCP2.6 | SSP5 RCP8.5 | SSP4 RCP3.4 | SSP4 RCP6.0 | SSP3 RCP7.0 | SSP2 RCP4.5 |
|---|---|---|---|---|---|---|---|---|---|---|
| **Transitions** | | | | | | | | | | |
| Total gross transitions | $10^6$ km$^2$ yr$^{-1}$ | 2100 | 2.02 | 3.99 | 2.12 | 4.21 | 4.56 | 4.79 | 4.60 | 3.06 |
| Total net transitions | $10^6$ km$^2$ yr$^{-1}$ | 2100 | 0.02 | 0.04 | -0.11 | 0.03 | 0.16 | 0.09 | 0.13 | 0.03 |
| **Human land use impacts** | | | | | | | | | | |
| Secondary land increase that is forested | % | 2015-2100 | 49.7 | 54.1 | 48.9 | 58.4 | 60.0 | 71.6 | 63.6 | 72.8 |
| U.S. Forests that are secondary | % | 2100 | 100.0 | 100.0 | 100.0 | 100.0 | 100.0 | 100.0 | 100.0 | 100.0 |
| Global area covered by natural vegetation in biodiversity hotspots | % | 2100 | 1.1 | 0.9 | 1.1 | 0.9 | 0.6 | 0.8 | 0.9 | 0.9 |
| Median secondary forest mean age | yr | 2100 | 74.0 | 58.5 | 74.2 | 67.7 | 60.8 | 60.6 | 68.0 | 63.0 |
| Land impacted by human land use | % | 2100 | 68.6 | 70.2 | 68.6 | 71.4 | 75.4 | 74.1 | 73.3 | 71.9 |
| Secondary land increase | $10^6$ km$^2$ | 2100-2015 | 13 | 10 | 13 | 12 | 6 | 10 | 8 | 12 |
| Secondary land increase (forest) | $10^6$ km$^2$ | 2100-2015 | 6 | 5 | 6 | 7 | 4 | 7 | 5 | 8 |
| Secondary land increase (non-forest) | $10^6$ km$^2$ | 2100-2015 | 7 | 5 | 7 | 5 | 2 | 3 | 3 | 3 |
| **Wood harvest and agricultural clearing** | | | | | | | | | | |
| Wood clearing for crop and pasture | Pg C | 2100-2015 | 47 | 56 | 47 | 47 | 88 | 59 | 70 | 44 |
| Total wood harvest | Pg C | 2100-2015 | 93 | 139 | 95 | 141 | 145 | 148 | 131 | 139 |
| Direct wood harvest | Pg C | 2100-2015 | 93 | 139 | 95 | 141 | 145 | 148 | 131 | 139 |
| Agricultural clearing for wood harvest | Pg C | 2100-2015 | 0 | 0 | 0 | 0 | 0 | 0 | 0 | 0 |
| **Shifting cultivation** | | | | | | | | | | |
| Agricultural land for shifting cultivation | $10^6$ km$^2$ yr$^{-1}$ | 2100 | 0 | 0 | 0 | 0 | 0 | 0 | 0 | 0 |
| **Forest loss and area** | | | | | | | | | | |
| Forest area change | $10^6$ km$^2$ | 2100-2015 | 0.9 | -1.3 | 0.9 | -0.9 | -5.1 | -1.4 | -3.4 | 0.8 |
| Forest area | $10^6$ km$^2$ | 2100 | 38.1 | 35.9 | 38.1 | 36.3 | 32.1 | 35.8 | 33.8 | 38.0 |
| Forest loss | $10^6$ km$^2$ | 2015-2100 | 12.0 | 17.6 | 12.1 | 15.3 | 20.3 | 17.9 | 15.1 | 15.0 |
| **Management** | | | | | | | | | | |
| Fuelwood | Pg C | 2100 | 0.2 | 0.7 | 0.2 | 0.9 | 0.9 | 0.9 | 0.8 | 0.7 |
| Wood-harvest | Pg C | 2100 | 0.9 | 1.6 | 0.9 | 1.7 | 1.8 | 1.9 | 1.5 | 1.5 |
| Fertilizer use | Tg N yr$^{-1}$ | 2100 | 140 | 223 | 177 | 110 | 240 | 145 | 173 | 210 |
| Irrigated area | $10^6$ km$^2$ | 2100 | 2.9 | 2.8 | 2.9 | 3.4 | 2.7 | 2.7 | 4.1 | 2.6 |
| Flooded area | $10^6$ km$^2$ | 2100 | 0.9 | 0.2 | 0.9 | 0.6 | 0.8 | 0.9 | 0.9 | 1.0 |
| Biofuel area | $10^6$ km$^2$ | 2100 | 3.6 | 10.9 | 3.4 | 0.2 | 18.0 | 3.7 | 0.0 | 0.0 |
| **Biomass** | | | | | | | | | | |
| Plant total biomass on all lands | Pg C | 2100 | 433 | 380 | 434 | 386 | 319 | 367 | 355 | 401 |



| Plant AGB on pantropical forest lands | Pg C | 2100 | 239 | 217 | 239 | 213 | 170 | 198 | 178 | 221 |
|---|---|---|---|---|---|---|---|---|---|---|---|
| Plant total biomass on forest lands | Pg C | 2100 | 390 | 343 | 391 | 349 | 290 | 335 | 322 | 366 |





Table 6. Regional results for 1700-2000 (historical period).

| | Secondary area (10⁶ km²) | Secondary Age (yr) | Gross Transitions (10³ km² yr⁻¹) | Net Transitions (10³ km² yr⁻¹) |
|---|---|---|---|---|
| 1700-1799 mean | | | | |
| North America | 0.3 | 150 | 12 | 3 |
| South America | 0.3 | 77 | 40 | 1 |
| Eurasia | 8.5 | 429 | 456 | 41 |
| Africa | 6.0 | 245 | 165 | 10 |
| Oceania | 0.1 | 98 | 5 | 1 |
| | | | | |
| 1800-1899 mean | | | | |
| North America | 0.3 | 144 | 52 | 33 |
| South America | 0.4 | 79 | 61 | 11 |
| Eurasia | 9.8 | 377 | 660 | 76 |
| Africa | 6.5 | 257 | 191 | 19 |
| Oceania | 0.1 | 116 | 13 | 10 |
| | | | | |
| 1900-1999 mean | | | | |
| North America | 1.7 | 52 | 108 | 48 |
| South America | 0.8 | 53 | 145 | 48 |
| Eurasia | 12.4 | 289 | 604 | 121 |
| Africa | 6.8 | 232 | 404 | 80 |
| Oceania | 0.1 | 99 | 40 | 33 |





Table 7. Regional results averaged over years 2000-2099.

| | Secondary area (10⁶ km²) | Secondary Age (yr) | Gross Transitions (10³ km² yr⁻¹) | Net Transitions (10³ km² yr⁻¹) |
|---|---|---|---|---|
| **SSP1 RCP1.9** | | | | |
| North America | 4.5 | 64 | 89 | 4 |
| South America | 2.5 | 46 | 129 | 9 |
| Eurasia | 18.4 | 210 | 1084 | 13 |
| Africa | 10.9 | 77 | 959 | 35 |
| Oceania | 0.9 | 46 | 20 | -4 |
| **SSP1 RCP2.6** | | | | |
| North America | 4.4 | 65 | 86 | 6 |
| South America | 2.5 | 47 | 128 | 9 |
| Eurasia | 18.2 | 213 | 1073 | 19 |
| Africa | 10.9 | 76 | 975 | 34 |
| Oceania | 0.9 | 48 | 18 | -4 |
| **SSP4 RCP3.4** | | | | |
| North America | 4.1 | 60 | 153 | 19 |
| South America | 3.0 | 45 | 109 | -3 |
| Eurasia | 17.1 | 197 | 1794 | 93 |
| Africa | 9.2 | 69 | 1629 | 143 |
| Oceania | 0.8 | 48 | 21 | 1 |
| **SSP5 RCP3.4OS** | | | | |
| North America | 4.0 | 62 | 171 | 15 |
| South America | 2.0 | 49 | 135 | 16 |
| Eurasia | 17.8 | 195 | 1936 | 50 |
| Africa | 10.6 | 81 | 798 | 49 |
| Oceania | 0.8 | 49 | 18 | -3 |
| **SSP2 RCP4.5** | | | | |
| North America | 4.2 | 65 | 92 | 7 |
| South America | 2.3 | 45 | 147 | 13 |
| Eurasia | 17.7 | 206 | 1379 | 44 |
| Africa | 10.9 | 69 | 1337 | 71 |
| Oceania | 0.8 | 49 | 20 | -4 |
| **SSP4 RCP6.0** | | | | |
| North America | 4.1 | 63 | 107 | 12 |
| South America | 2.4 | 45 | 130 | 3 |
| Eurasia | 17.9 | 201 | 1752 | 53 |
| Africa | 9.5 | 64 | 1607 | 133 |
| Oceania | 0.7 | 50 | 18 | -2 |
| **SSP3 RCP7.0** | | | | |
| North America | 3.8 | 66 | 94 | 17 |
| South America | 2.0 | 49 | 132 | 24 |
| Eurasia | 18.1 | 208 | 1449 | 32 |
| Africa | 9.5 | 70 | 1877 | 133 |
| Oceania | 0.7 | 53 | 16 | 1 |
| **SSP5 RCP8.5** | | | | |
| North America | 4.0 | 67 | 81 | 15 |
| South America | 2.1 | 49 | 126 | 19 |
| Eurasia | 17.7 | 209 | 1594 | 48 |
| Africa | 10.8 | 70 | 1539 | 62 |
| Oceania | 0.9 | 50 | 16 | -4 |






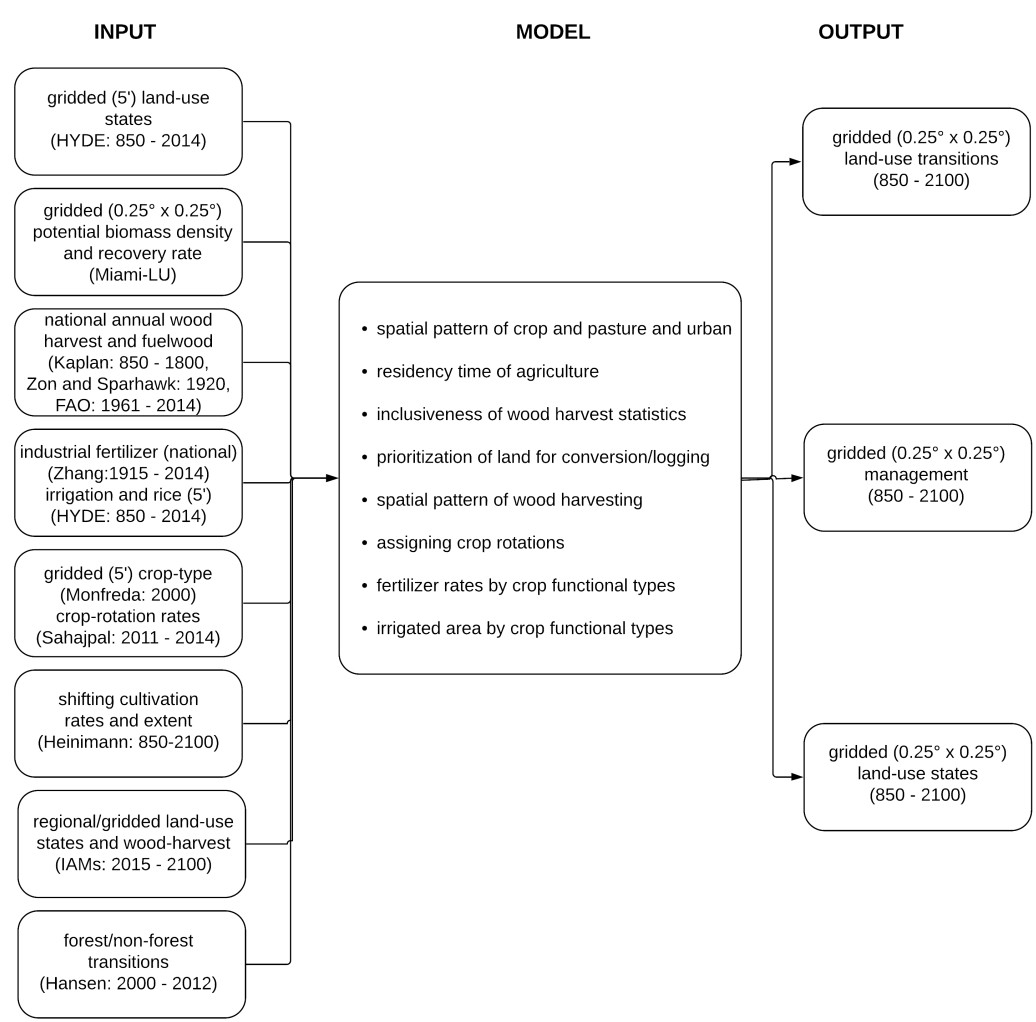

Figure 1. Schematic diagram of major model inputs, decisions, and outputs.





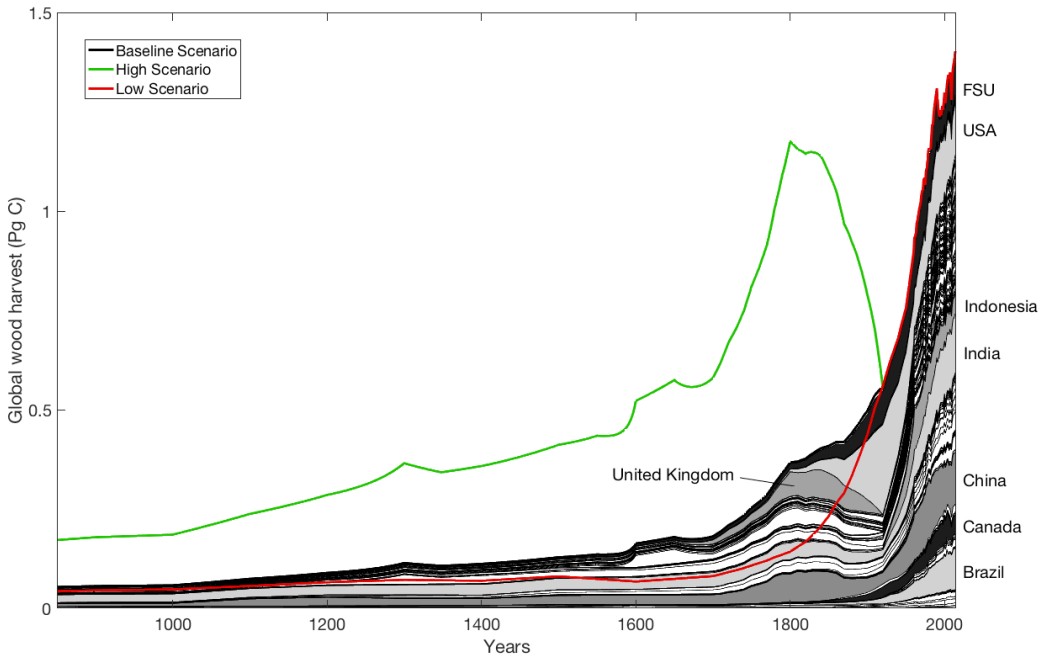

Figgure 2. (a) Annual national wood harvest (in Pg C/y) for 850-2015, for low, baseline and high scenarios. (FSU= Former Soviet Union.) Integrated total wood harvest in baseline scenario was 259 PgC (including slash).






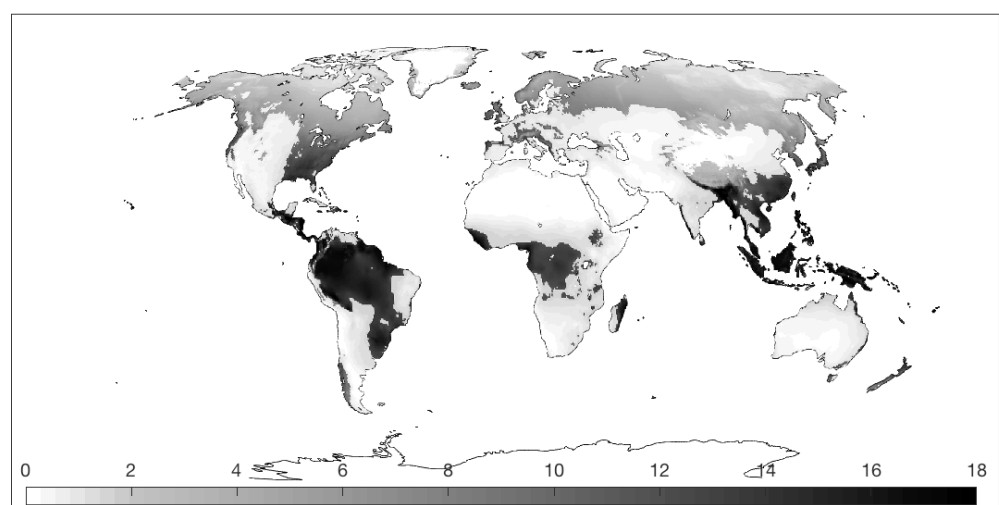

Figure 3. Global potential above-ground biomass (kg C m$^2$) as estimated by Miami-LU model. Land is considered to be potential forest if the potential biomass density is >2 kg C m$^{-2}$ (after Hurtt et al., 2006; 2011).



Figure 4. Pre-harmonization (a) global cropland, (b) global grazing land, (c) 0.25° grid cell comparison of 2015 crop fraction of grid cell areas (excluding water and ice): LUH2 (x-axis), IAM (y-axis).







Figure 5. Post-harmonization comparison of projected changes 2015-2100 at multiple scales (0.25 degree, 2 degree, regional), as fraction of total area. Original IAM change (x-axis), harmonized change (y-axis), for (a) Cropland, and (b) grazing land. Note that for SSP4 RCP3.4, SSP2 RCP4.5, and SSP4 RCP6.0, pasture was only reported by IAMs as regional totals, so LHU2 comparisons at 0.25° and 2° are not possible.





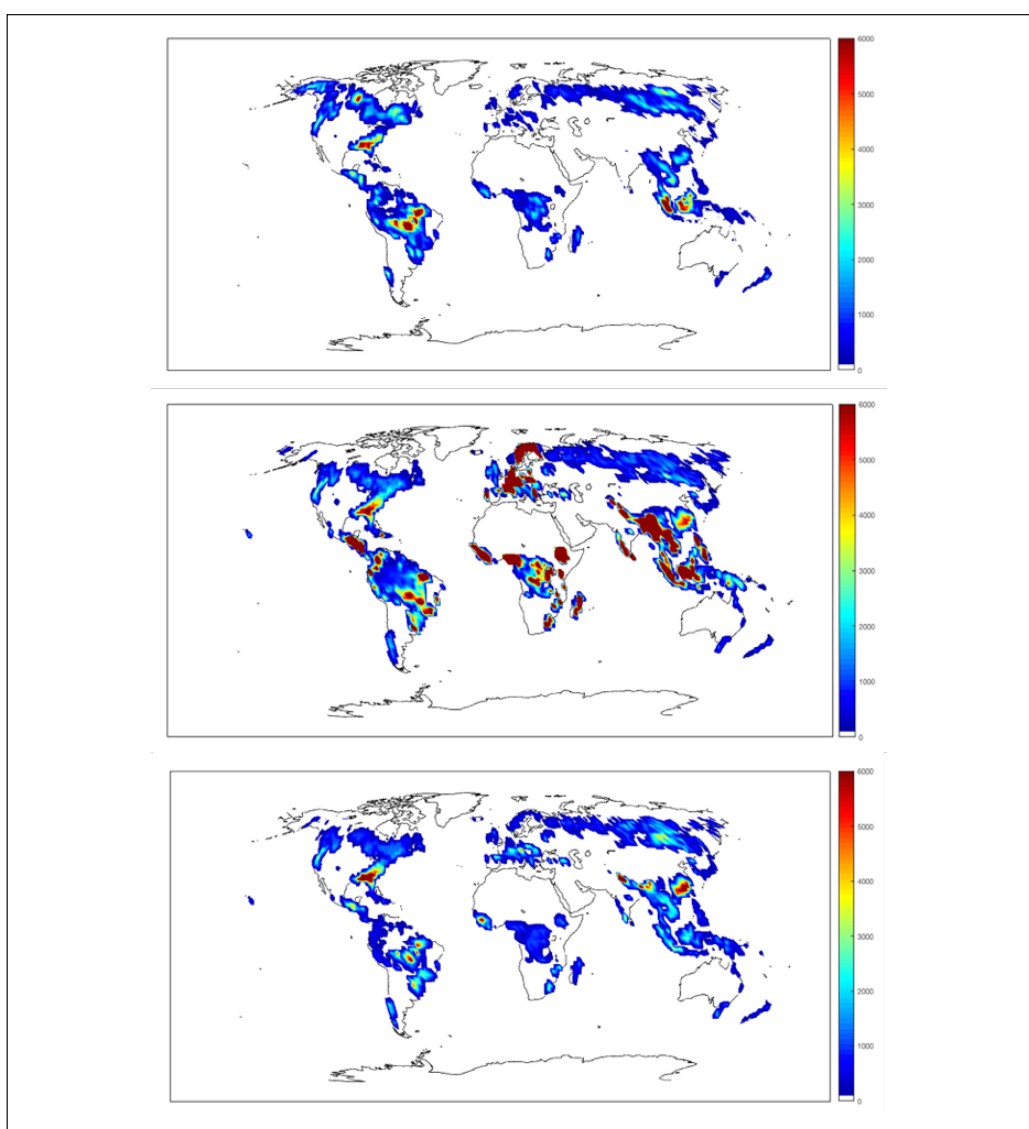

Figure 6. Forest loss 2000-2012 (a) Landsat forest loss (Hansen et al. 2013), (b) LUH2 forest loss without Landsat constraint, (c) LUH2 forest loss with Landsat constraint.




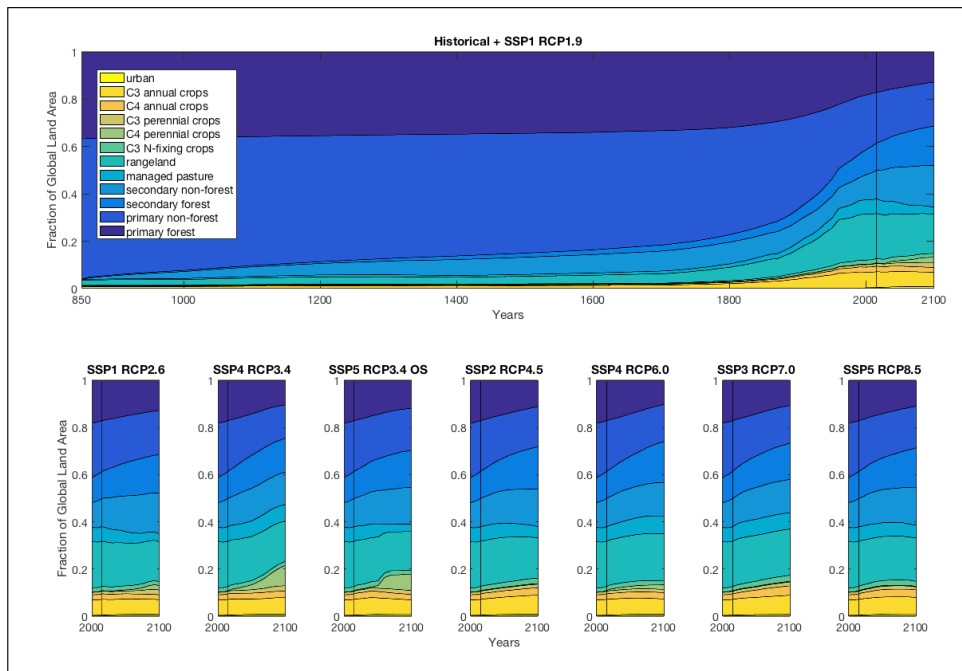

Figure 7. Harmonized global land-use area fractions 850-2015 (baseline historical) and 2015-2100 for the 8 future scenarios.


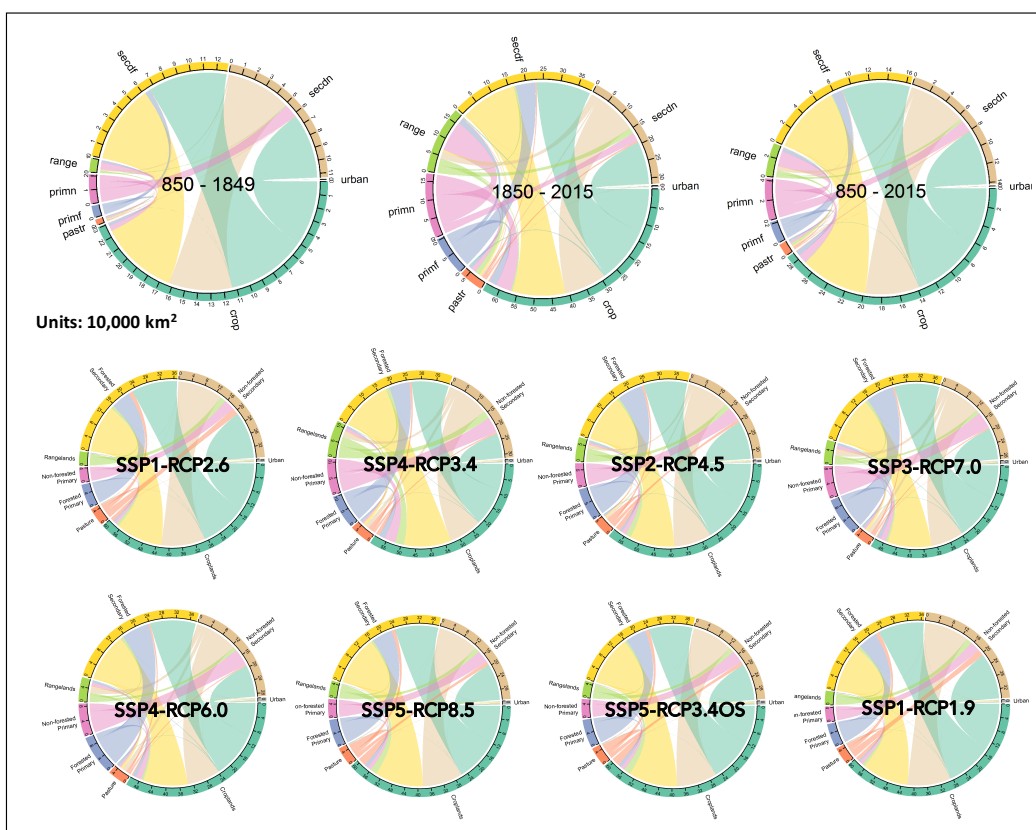

Figure 8. Global land-use transitions by time-period and by future scenario.

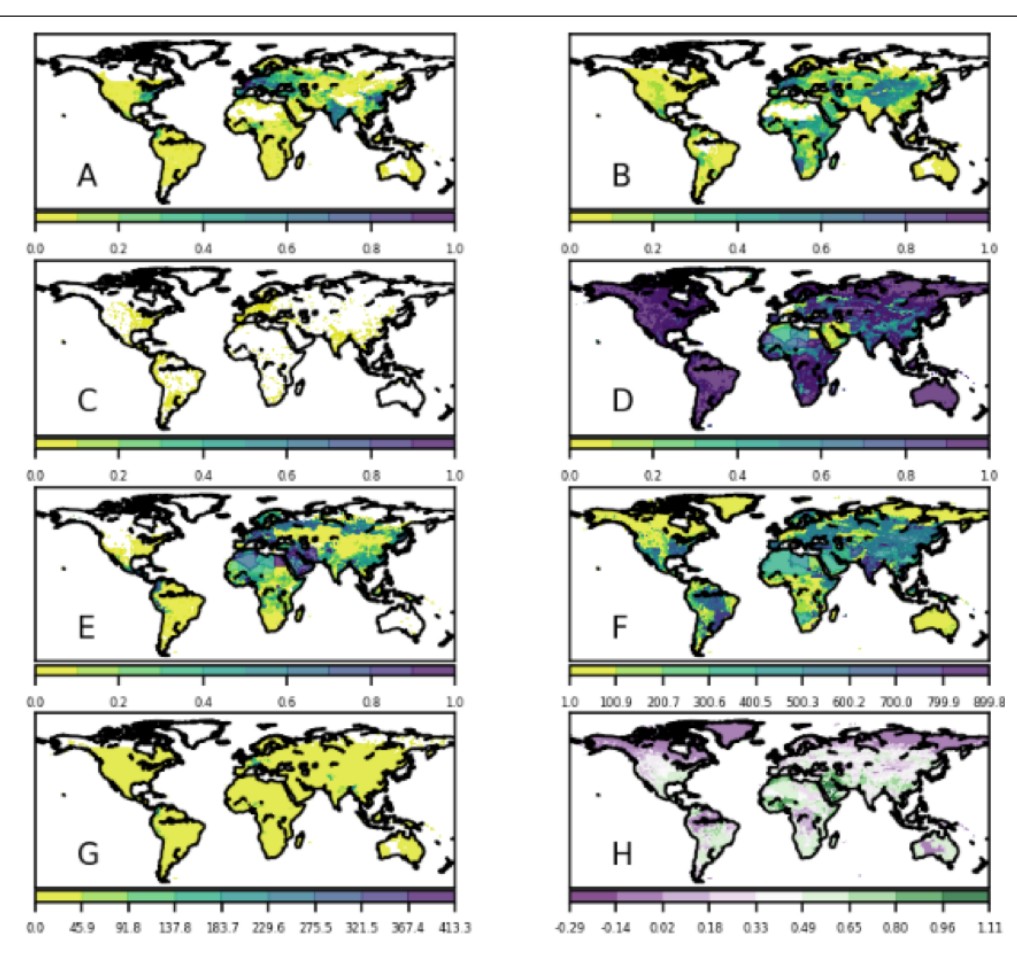

Figure 9. Maps for year 1850 showing **a**. fraction of each grid cell occupied by cropland **b**. fraction of each grid cell occupied by pasture, **c**. fraction of each grid cell occupied by urban land, **d**. fraction of each grid cell occupied by primary vegetation, **e**. fraction of each grid cell occupied by secondary vegetation, **f**. mean age (in years) of secondary lands in each half degree grid cell, **g**. mean gross transitions (km$^2$ year$^{-1}$) over 20 year interval for each grid cell, **h**. mean net transitions (km$^2$ year$^{-1}$) over 20 year interval for each grid cell.


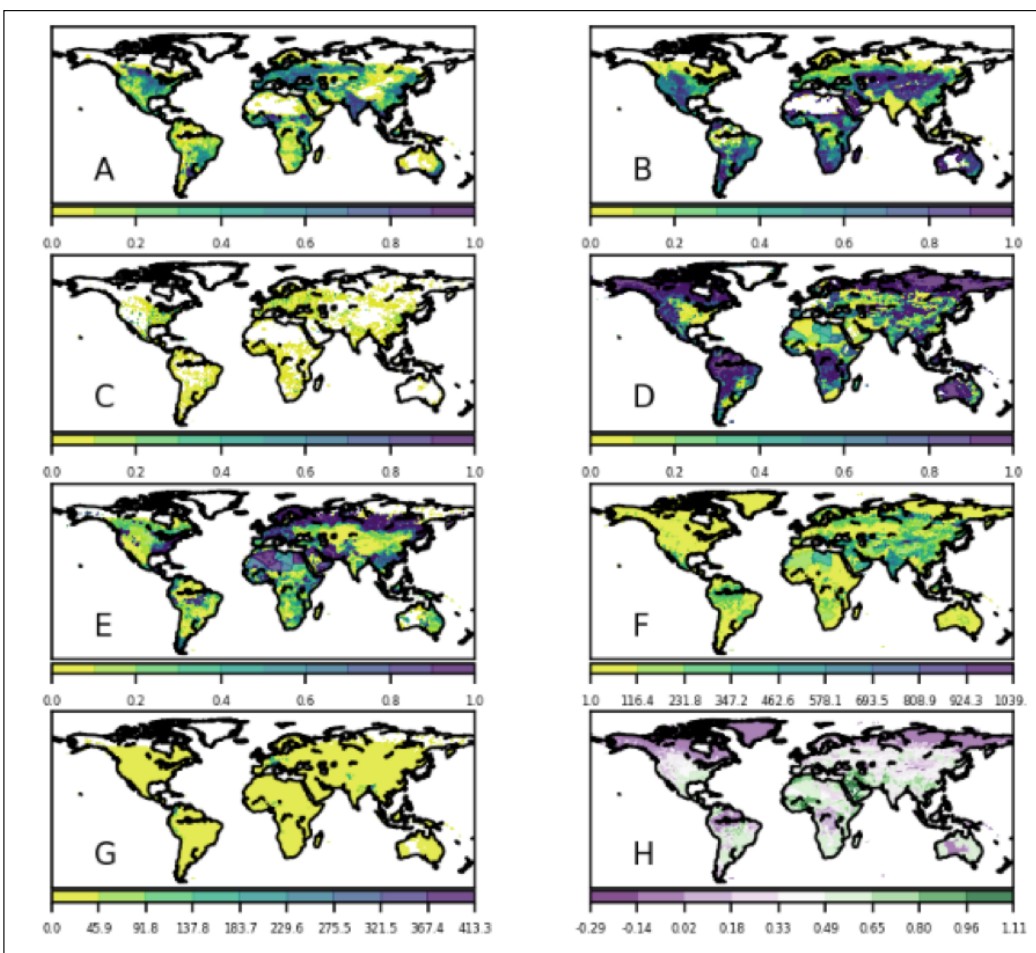

Figure 10. Maps for year 2015 showing **a**. fraction of each grid cell occupied by cropland **b**. fraction of each grid cell occupied by pasture, **c**. fraction of each grid cell occupied by urban land, **d**. fraction of each grid cell occupied by primary vegetation, **e**. fraction of each grid cell occupied by secondary vegetation, **f**. mean age (in years) of secondary lands in each half degree grid cell, **g**. mean gross transitions (km$^2$ year$^{-1}$) over 20 year interval for each grid cell, **h**. mean net transitions (km$^2$ year$^{-1}$) over 20 year interval for each grid cell.



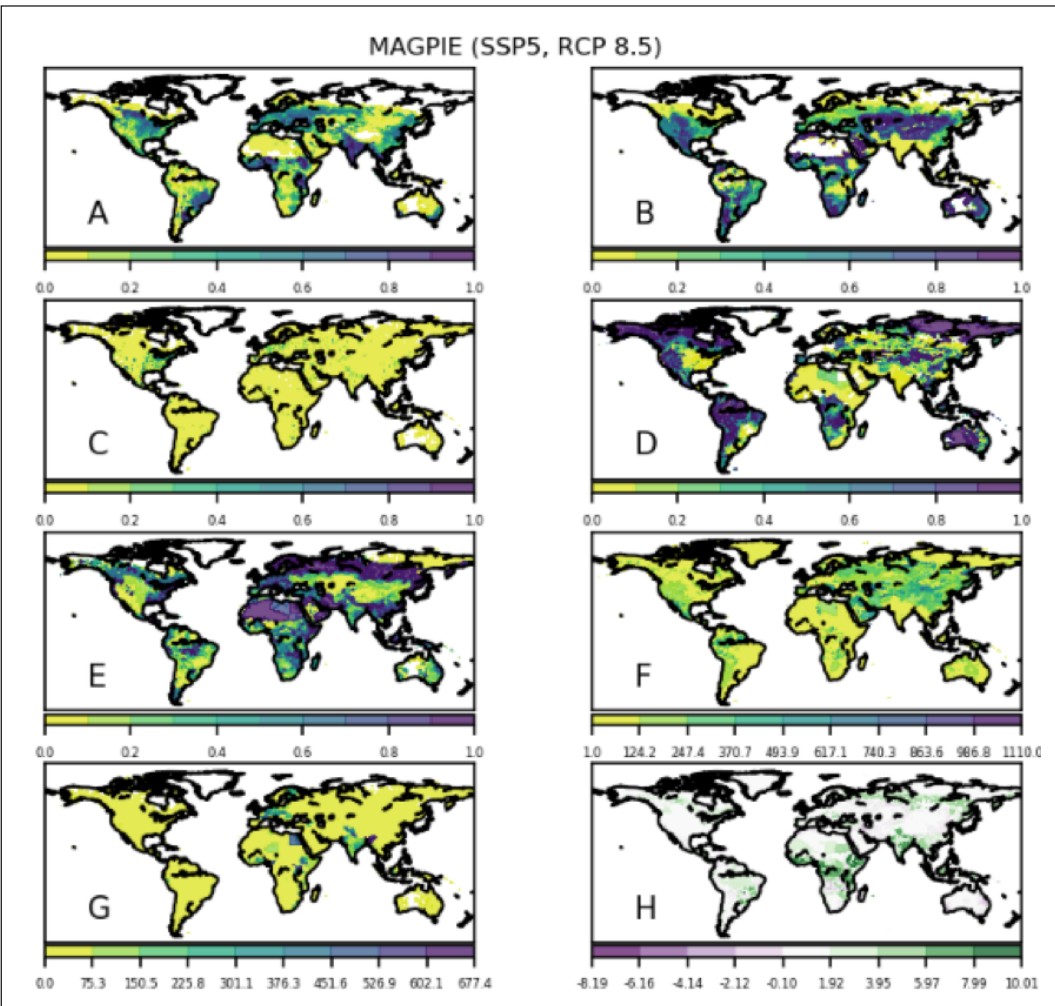

Figure 11. Maps for year 2100 for SSP5 RCP8.5 scenario showing **a**. fraction of each grid cell occupied by cropland **b**. fraction of each grid cell occupied by pasture, **c**. fraction of each grid cell occupied by urban land, **d**. fraction of each grid cell occupied by primary vegetation, **e**. fraction of each grid cell occupied by secondary vegetation, **f**. mean age (in years) of secondary lands in each half degree grid cell, **g**. mean gross transitions (km$^2$ year$^{-1}$) over 20 year interval for each grid cell, **h**. mean net transitions (km$^2$ year$^{-1}$) over 20 year interval for each grid cell.



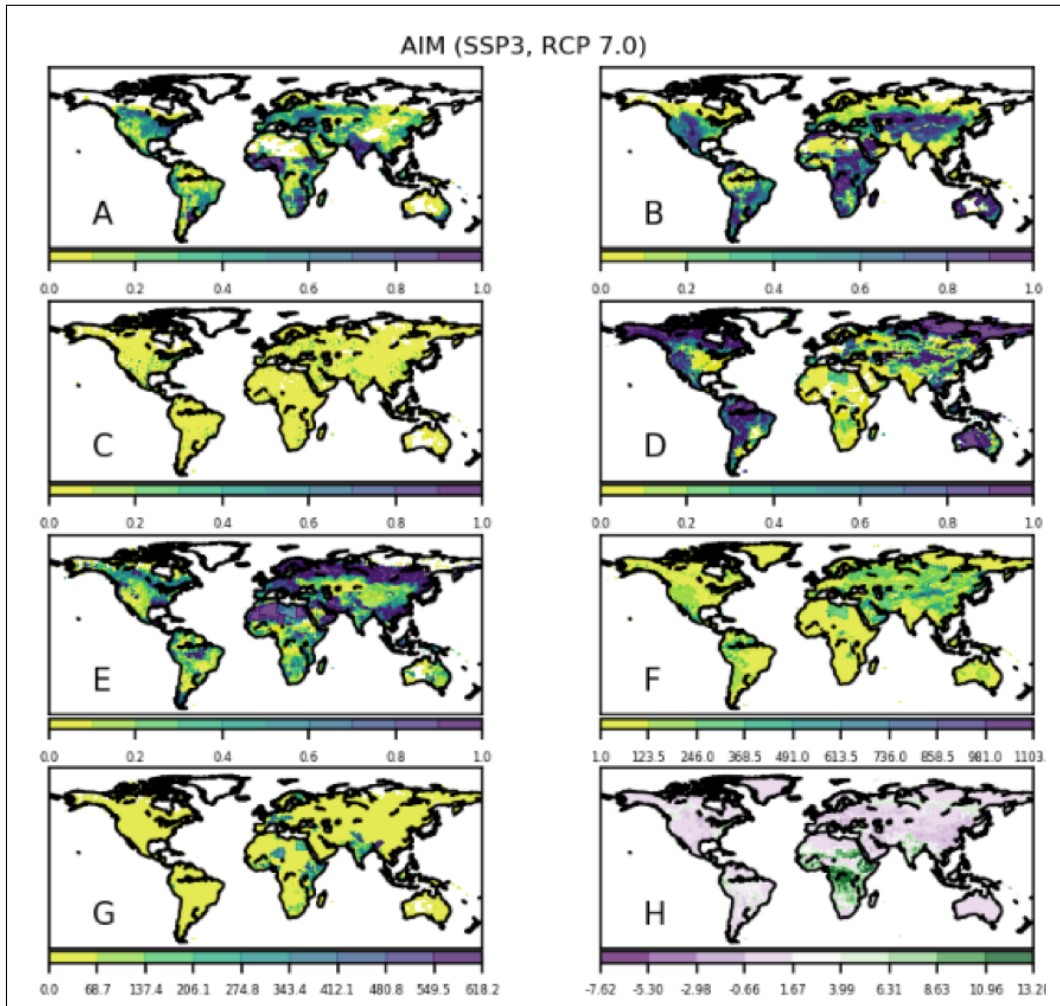

Figure 12. Maps for year 2100 for SSP3 RCP7.0 scenario showing **a**. fraction of each grid cell occupied by cropland **b**. fraction of each grid cell occupied by pasture, **c**. fraction of each grid cell occupied by urban land, **d**. fraction of each grid cell occupied by primary vegetation, **e**. fraction of each grid cell occupied by secondary vegetation, **f**. mean age (in years) of secondary lands in each half degree grid cell, **g**. mean gross transitions (km$^2$ year$^{-1}$) over 20 year interval for each grid cell, **h**. mean net transitions (km$^2$ year$^{-1}$) over 20 year interval for each grid cell.

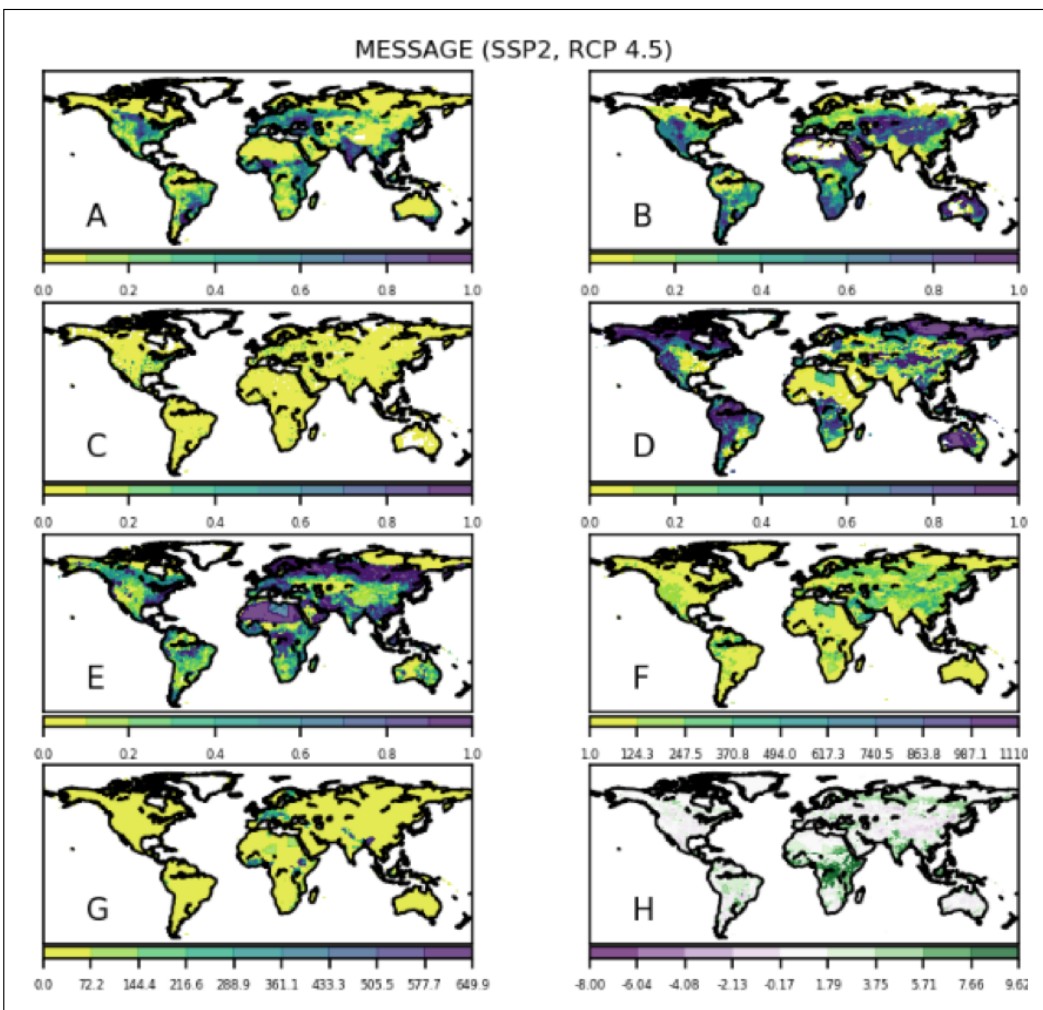

Figure 13. Maps for year 2100 for SSP2 RCP4.5 scenario showing **a**. fraction of each grid cell occupied by cropland **b**. fraction of each grid cell occupied by pasture, **c**. fraction of each grid cell occupied by urban land, **d**. fraction of each grid cell occupied by primary vegetation, **e**. fraction of each grid cell occupied by secondary vegetation, **f**. mean age (in years) of secondary lands in each half degree grid cell, **g**. mean gross transitions (km$^2$ year$^{-1}$) over 20 year interval for each grid cell, **h**. mean net transitions (km$^2$ year$^{-1}$) over 20 year interval for each grid cell.


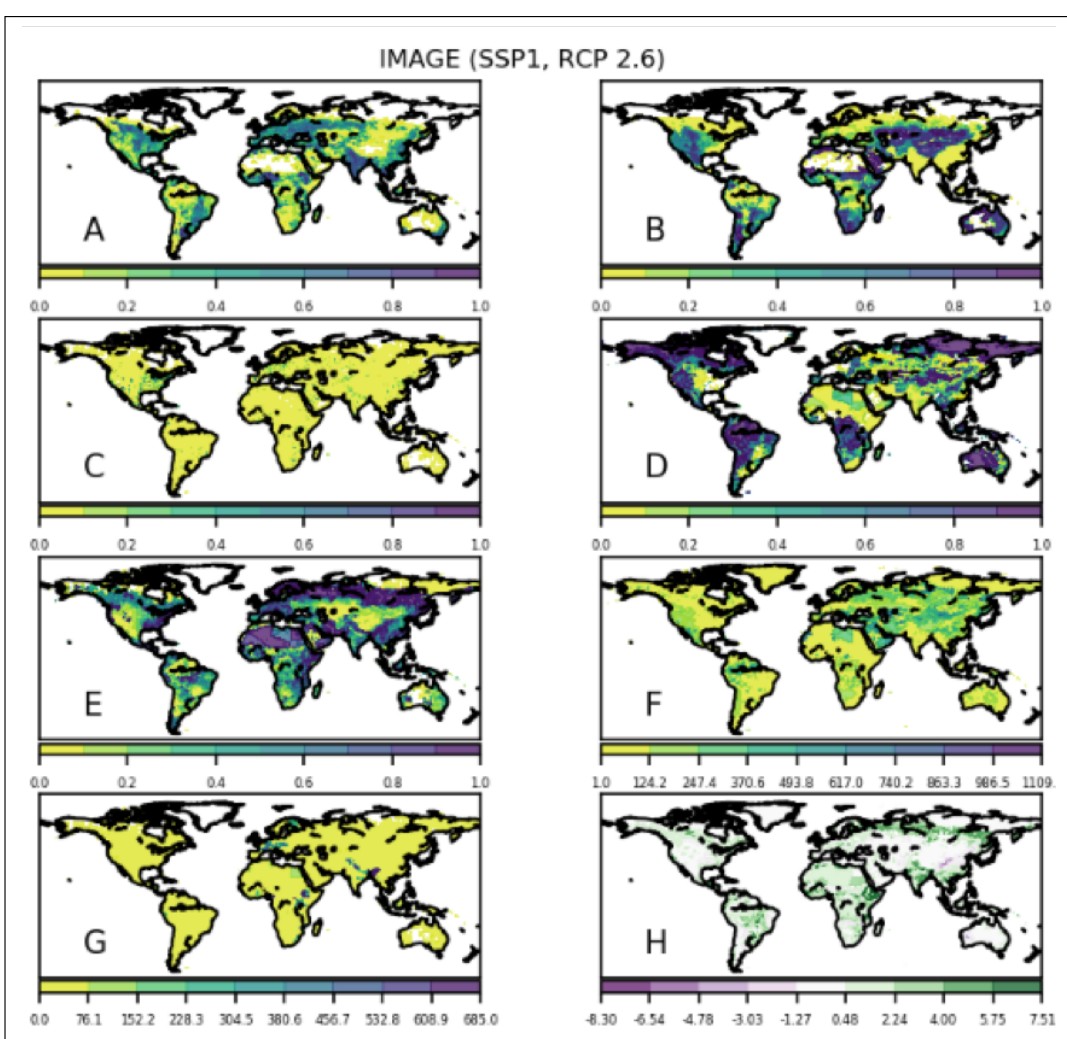

Figure 14. Maps for year 2100 for SSP1 RCP2.6 scenario showing **a**. fraction of each grid cell occupied by cropland **b**. fraction of each grid cell occupied by pasture, **c**. fraction of each grid cell occupied by urban land, **d**. fraction of each grid cell occupied by primary vegetation, **e**. fraction of each grid cell occupied by secondary vegetation, **f**. mean age (in years) of secondary lands in each half degree grid cell, **g**. mean gross transitions (km$^2$ year$^{-1}$) over 20 year interval for each grid cell, **h**. mean net transitions (km$^2$ year$^{-1}$) over 20 year interval for each grid cell.











Fig. 15. Time series of harmonized management variables.





**Appendix**

**Mapped patterns of Tier 2 Scenarios**

Figure A1. Maps for year 2100 for SSP4 RCP6.0 scenario showing **a**. fraction of each grid cell occupied by cropland **b**. fraction of each grid cell occupied by pasture, **c**. fraction of each grid cell occupied by urban land, **d**. fraction of each grid cell occupied by primary vegetation, **e**. fraction of each grid cell occupied by secondary vegetation, **f**. mean age (in years) of secondary lands in each half degree grid cell, **g**. mean gross transitions (km$^2$ year$^{-1}$) over 20 year interval for each grid cell, **h**. mean net transitions (km$^2$ year$^{-1}$) over 20 year interval for each grid cell.


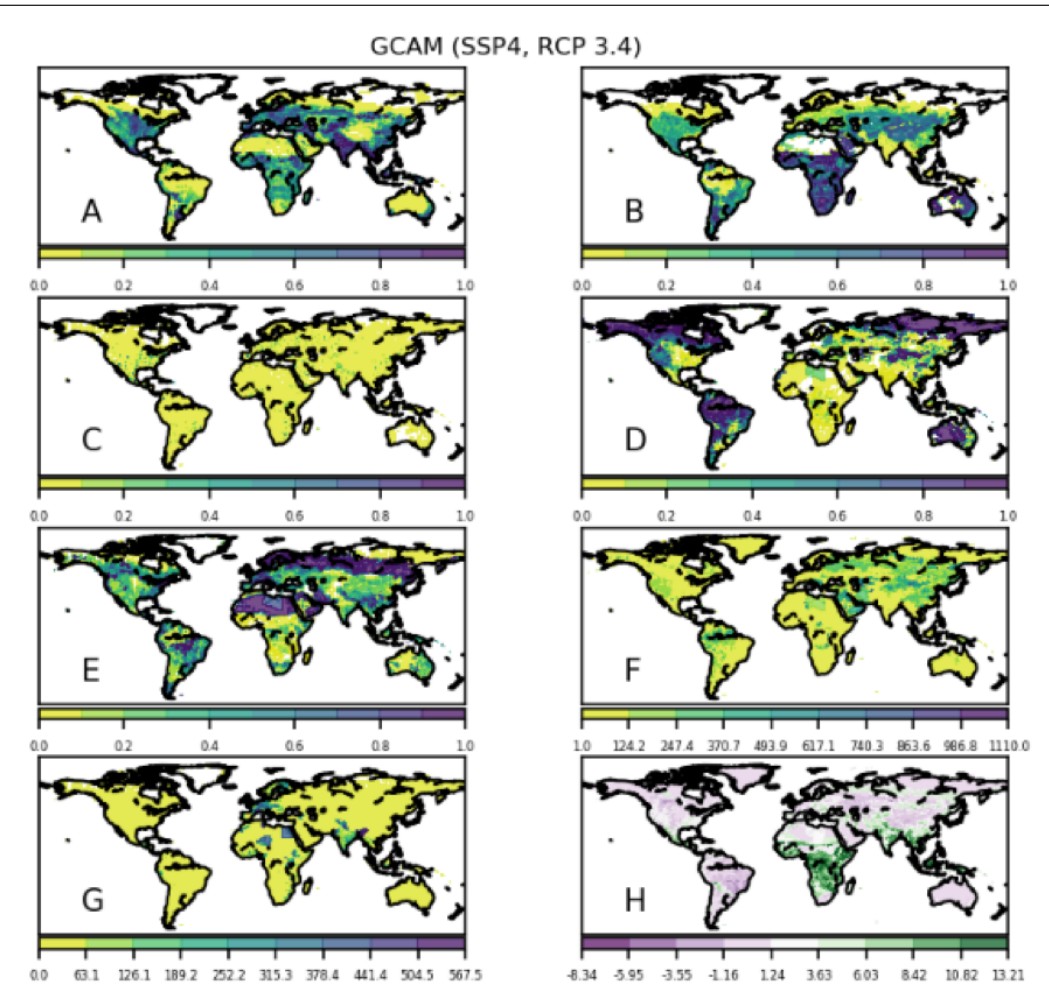

Figure A2. Maps for year 2100 for SSP4 RCP3.4 scenario showing **a**. fraction of each grid cell occupied by cropland **b**. fraction of each grid cell occupied by pasture, **c**. fraction of each grid cell occupied by urban land, **d**. fraction of each grid cell occupied by primary vegetation, **e**. fraction of each grid cell occupied by secondary vegetation, **f**. mean age (in years) of secondary lands in each half degree grid cell, **g**. mean gross transitions ($km^2$ $year^{-1}$) over 20 year interval for each grid cell, **h**. mean net transitions ($km^2$ $year^{-1}$) over 20 year interval for each grid cell.



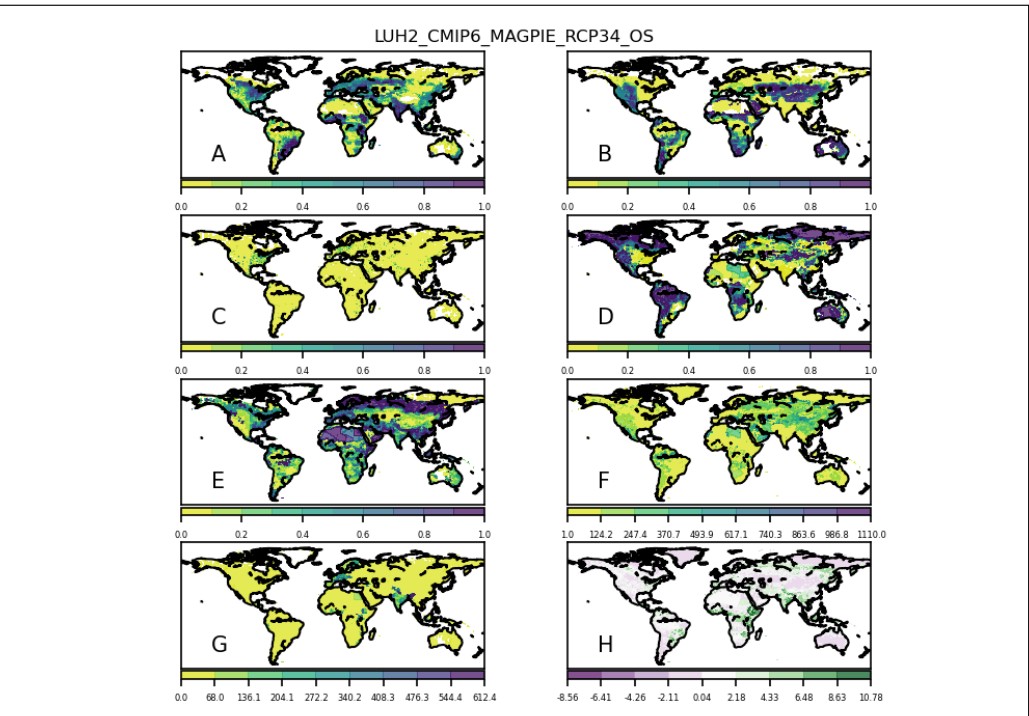

Figure A3. Maps for year 2100 for SSP5 RCP3.4OS scenario showing **a**. fraction of each grid cell occupied by cropland **b**. fraction of each grid cell occupied by pasture, **c**. fraction of each grid cell occupied by urban land, **d**. fraction of each grid cell occupied by primary vegetation, **e**. fraction of each grid cell occupied by secondary vegetation, **f**. mean age (in years) of secondary lands in each half degree grid cell, **g**. mean gross transitions ($km^2$ $year^{-1}$) over 20 year interval for each grid cell, **h**. mean net transitions ($km^2$ $year^{-1}$) over 20 year interval for each grid cell.



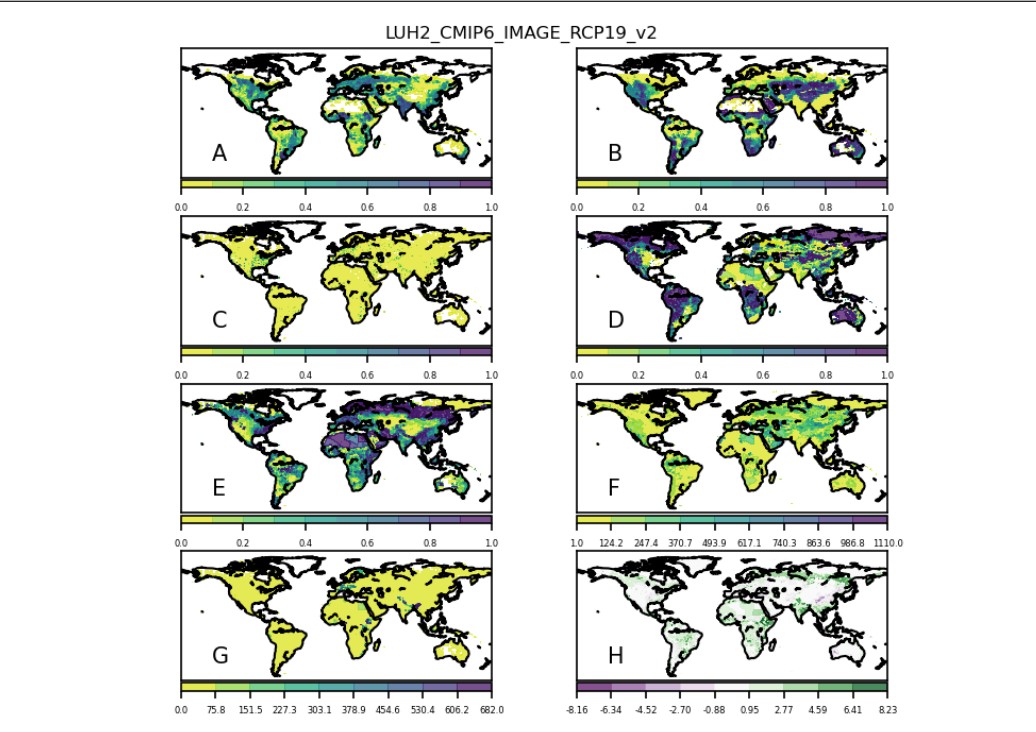

Figure A4. Maps for year 2100 for SSP1 RCP1.9 scenario showing **a**. fraction of each grid cell occupied by cropland **b**. fraction of each grid cell occupied by pasture, **c**. fraction of each grid cell occupied by urban land, **d**. fraction of each grid cell occupied by primary vegetation, **e**. fraction of each grid cell occupied by secondary vegetation, **f**. mean age (in years) of secondary lands in each half degree grid cell, **g**. mean gross transitions (km$^2$ year$^{-1}$) over 20 year interval for each grid cell, **h**. mean net transitions (km$^2$ year$^{-1}$) over 20 year interval for each grid cell.