# Peer review of "Harmonization of Global Land-Use Change and Management for the Period 850-2100 (LUH2) for CMIP6"

_Geoscientific Model Development, 2019_

## Referee Comment (RC1) · Anonymous Referee #1 · 4 Jun 2020

General comments:

The authors seek to provide fractional land-use patterns, underlying land-use transitions, and key agricultural management information for the time period 850-2100 at 0.25 degree resolution, all while having a smooth transition between methods applied to past and future results and preserving the future changes. Such an effort is of high importance to both the land surface and climate modelling communities, as maps of this type are essential drivers in simulations studying the feedbacks between the land surface and the climate.

This work builds on a previous product, LUH1. Some changes are an increase in

spatial resolution (0.25 degree instead of 0.5 degree); a larger list of 12 subgrid scale land-use types used in the land-use transition matrix; and tracking of key management activities related to agriculture. The authors clearly state assumptions they make in the work at various points in the manuscript, which is fine; I don't require justification for clearly-stated assumptions, although it's always welcome.

Overall the work is high quality, and I believe should be published with only minor revisions.

Specific comments:

It was only in section 2.12 that I understood that eight different LUH2 scenarios are available for the future, and (I assume) one for the historical period. I was under the impression that the IAM runs were somehow averaged to create one trajectory from 850-2100. Assuming I am not mistaken, it would be good to clarify that in the abstract and/or introduction, to let users know precisely how many trajectories are available.

In section 2, line 125, A and m are solved for. That implies that f and l in Eqs. 1 and 3 are already known. For someone unfamiliar with the process, it's not clear how the inputs listed in lines 127-133 lead to this, before reading all of the detailed description. A short sentence making the explicit link would be welcome here to ease the reader in to the more detailed discussion. For example, when discussing shifting cultivation (line 261), the authors "abandoned the Heinimann et al. (2017) prescribed percentage of total cropland area in the grid cell (e.g. cropland to secondary land), and cleared the same area from natural vegetation". This sounds like it could be A given in Eq 1. Therefore, it's not clear to me what is being solved for in Eqs. 1 and 3, and how the datasets listed in Section 2 contribute to that. Perhaps even a short paragraph would be useful to clear that up. Section 2.11.1 addresses this, but it's easy for the reader to be lost by then.

The wood harvest numbers are detailed in Section 2.5; however, it's not clear how this feeds into Eqs. 1 and 3. Combined with the forest loss map from 2.6? Perhaps this is

described a bit more in Sections 2.10.1 and 2.11.2, but it would be good to keep the reader on track earlier, making sure each section is tied to the overall goal of solving Eqs. 1 and 3.

From Section 2.9, I understand that LUH2 does not simply use output from IAM intercomparison projects or separate runs; significant work was carried out to harmonize the input and create a consistent set of IAM simulations specifically for this project. That deserves more emphasis, in my mind; incorporating a couple words around lines 90 would be good.

Is the paragraph starting around line 525 part of the "Harmonizing inputs" section? It seems to be part of a section dedicated to smoothing the transition between the historical and future datasets, although I don't seen such a dedicated section. I understand "Harmonizing inputs" to be harmonizing the data that goes into the IAM runs. If this paragraph is really discussing that, a sentence to make it more clear would be appreciated. Conversely, if "Harmonizing inputs" is completely dedicated to smoothing the transition between historical and future datasets, I would propose to make that more explicit, as the previous language never connected the idea of "Harmonizing inputs" to "Smoothing the transition".

In line 550, no rationale is given for holding the values between 0 to 500. I propose to either explicitly call this an assumption, or to add a sentence/reference giving the reason for these limits (well, the upper limit...limiting fertilizer application to positive numbers makes perfect sense, and thus the lower limit does not need justification). The following sentence could either apply to this choice of 0 to 500 or to the overall method described in the paragraph, and so a couple words to remove ambiguity would be welcome.

Figure 1 does a nice job of showing the inputs, model, and outputs. Perhaps a similar chart separated into historical maps, future maps, and the transition would also help the reader?

The paragraph beginning in line 707 is quite interesting. As I understand that the eight future trajectories were run specifically for this project, it would be nice to develop this further and see if some of the observations can be explained. For example, the idea that global cropland area and wood harvesting increased did not surprise me (increased population, increased food supply, wood/bioenergy products increasingly viewed as favorable for the environment). Six scenarios projecting decreased grazing land did surprise me, though, as I would have expected increased meat consumption as more societies shift to the "Western" meat-heavy diet. If the IAM scenarios had been taken from another source, it would not be necessary to explore them further here; however, as it seems that much work went into creating these trajectories for this project, I would appreciate another paragraph dedicated to this. I note that further observations are mentioned in later sections (Sec 3.3, for example), but I don't see any driving factors for the differences.

The badge plots in Figure 8 are great to depict the transition information. I do struggle with the font, though. Could the text indicating the land use (e.g., pasture) be instead put in a legend across the bottom, with the text in the same color as the sector and made larger, at least for the eight trajectories where the font is the smallest (it appears they share the same sectors)?

The authors claim the new LUH2 products contain fifty times the information as the previous LUH1 product on two separate occasions in prominent places (abstract and conclusion), but do not support this anywhere in the text. It would be best to either add supporting evidence for this claim or remove it.

The evaluation of the LUH2 dataset against diagnostics is most welcome, and it's particularly interesting to see the comparison also against LUH1 for these same diagnostics (Table 3). The diagnostics are not discussed in any detail, though, which makes me wonder how they are "new", as mentioned in the abstract. I would modify the abstract to remove the word "new" or add more discussion on the diagnostics themselves demonstrating that they are, in fact, new, why they were selected, and how tables 3

and 5 compare in their selected diagnostics (they are not identical, for some reason). Additionally, I would be much more interested to read in the abstract what the result of the assessment was rather than just reading that the assessment was done.

Some technical comments:

Line 81: "Le Quéré et al. ,2016" should be "Le Quéré et al., 2016"

Line 771: There is an open paranthesis, but no close in the next line. "reference value of 0.3 X 10 6 km2)."

Line 780: Sometimes petagrams of carbon is written "Pg C" in this paragraph; other-times "PgC". I prefer "Pg C".

Line 791: Four significant figures in these numbers seems pretty optimistic, as that implies an uncertainty of less than 0.1%. I would stick to three at the most (e.g., 13.3%), since this paragraph is reporting differences in the created datasets and not a real uncertainty compared to observations (I'm sure the uncertainty compared to real historical values would be fairly high). Some consistency should be applied to the number of significant digits reported in this paragraph (and likely the whole paper...I see anywhere from two to five for various numbers, with no clear reason).

Line 793: There is a stray "." after 58.

Line 812: The "yr -1" has been split over two lines.

Line 910: Uses "bookeeping". Next line using "bookkeeping". Line 596 uses "book-keeping".

---

## Referee Comment (RC2) · Anonymous Referee #2 · 6 Jun 2020

Review for gmd-2019-360:

In their manuscript "Harmonization of Global Land-Use Change and Management for the Period 850-2100 (LUH2) for CMIP6", Hurtt et al. present the methods used to create the LUH2 dataset – the detailed land cover maps used by the CMIP6 land surface models. The paper is well written and thorough, though it reads like a technical description (which is actually appropriate given the detailed methods required to understand the construction of the LUH2 dataset). I have only minor comments on the manuscript, and believe it is appropriate for publication in GMD.

Comments:

Line 50 – since the CMIP6 runs have already mostly happened, mentioning that here, and that this is a description/analysis of the land use patterns used by those simulations, would be appropriate (the authors mention it in the discussion).

Line 64 – typo: provided -> provide

Line 100 – I was confused about what GLM2 was, and what role it plays in LUH2. A bit of elaboration here on that would be helpful.

Line 115/116 – drop "i.e."

Line 120 – (equation 3) Please elaborate more on what equation 3 means. m = f, but the following sentence (lines 121-122) don't sound like m=f...

Line 155 - At this point, the reviewer wonder's "How are changes in ice and water fraction accounted for?" Are no gridcells that are 100% ice projected to be less than that in the future? Or do those gridcells get to be "bare ground" under the ice, and models with dynamic ice sheet models can convert to the regular land model as appropriate, but just use gravel/bare ground as the ground cover? Or is there a chance for grass to grow on melted glacial sites in 100-300 years? Maybe this dataset assumes land ice is fixed temporally – if so, please state. If not please elborate on how it is handled.

Line 203 - Is this because rice is the only C3 crop that is flooded? Is all rice in this dataset represented as flooded?

Line 370 – Section 2.8 provided a very clear and useful walk-through of the land use in each SSP – I thank the authors for laying this out so clearly.

Line 695 - "these 12 states" -> which 12 states? Maybe say "the 12 potential states of vegetation cover used in this dataset..." (remind/help the reader)

Line 728-730 - Elaborate on these three historical scenarios? (It feels a bit like they just showed up - if they were already introduced above it got lost in the details, and reminding the reader of what they are here would be helpful...)

Line 740 – "Badge plots": This is a minor comment, feel free to ignore it. I've never heard the term "badge plot" - I knew what it was when I looked at it, but have always heard of and referred to these plots as "circle network graphs" or "circle network diagrams". Putting "badge plot" into google didn't pop up any figures like this.

Line 760-763 – This is quite interesting! At this point, I wanted to hear more about it. The authors actually discuss it more later.

Line 785 – Typo: "above- ground" -> "above-ground" (remove space after dash)

Line 788-789 – as mentioned above, it wasn't entirely clear what the lower, baseline, and upper historical scenarios were.

Line 814 – Is this the same metric as used on Line 760? (If so, consider moving the two discussions to the same place - when I hit line 760 I wanted to know more about what it used to look like.)

Line 824 - tied to the above comment - secondary forest age is going up by quite a bit in RCP 8.5?

Line 936 - This is the list of datasets for Ma et al 2019 - maybe it is the same list, in which case no modification is necessary. If there is a separate list for this manuscript, then this link needs to be updated.

Figure 4 – Why the discrete drop in pasture-land after 2000? Cropland has a smoother transition from historical to projected land area.

Figure 5 – text is extremely small in subplot legends. Had to blow it up on the computer, and wouldn't be able to read it in print.

Figure 8 – as in Figure 5, text is too small to read

Figure 9-14 – are these images blurry?

Figure 15 – legend text too tiny to read

---

## Short Comment (SC1) · 9 Jun 2020

The paper by Hurtt et al. describes the LUH2 dataset and the harmonisation required to achieve a continuous dataset of human land-use activities required for CMIP6 ESM simulations. Thank you for this tremendous effort and this dataset.

While the scope of the methods and the results seem to cover the important aspects, I would appreciate if the discussion section would be a bit more elaborate. Topics that could be helpful to be discussed would be

(a) in particular a discussion of emerged/possible issues for the usage of the data in LSMs/ESMs, since this is stated to be one of the main goals of the dataset (three examples coming into my mind are (1) how to deal with the interpolations used to construct wood harvest; (2) how to deal with inconsistencies between the static forest/non-forest-map used in LUH2 when calculating with another (disagreeing) vegetation cover; (3)how to treat rangelands);

(b) a discussion of uncertainties (including the three scenarios but if possible even broader; further examples could be pointing to where uncertainties of underlying data are discussed; conflicts of assumptions taken with shifts in country borders; climate dependent changes in biomass (or is this accounted for in the MIAMI-LU model?),...);

(c) comparisons to other datasets, e.g. the land cover dataset ESA-CCI?

E.g. regarding (1) the interpolations used to construct wood harvest: the two time dependent interpolation rules (regarding slash fractions and inclusiveness of wood cut in conversions in the wood harvest statistics) probably cause problems for a straightforward usage of the data in ESMs. The fate of the carbon in the LSMs matter, i.e. it is important to which carbon pool the harvested carbon should be assigned (such as construction wood, fire wood, etc, since these have different turnover times).

(2) albeit LUH2 is a land-use data set and not a land cover dataset, there are several assumptions where information on the land cover is used in the construction of the LUH2 data and which might be in conflict with the land cover assumed in an ESM simulation?

(3) there has been quite some discussion about this in the TRENDY process, particularly if rangelands involve land cover change, or not.

Furthermore, while reading I sometimes had the feeling that knowledge required for understanding was only given later (e.g. how grazing lands are treated and when rangelands and pastures are aggregated for grazing lands and information about the uncertainty scenarios).

Some specific comments:

l.80: Starting the 2016 version of the global carbon project (GCB) LUH2 has been used, i.e. LUH1 has not been used for the following GCBs: Le Quéré et al., 2016; Le Quéré et al., 2017; Le Quéré et al., 2018; Friedlingstein et al., 2019

l.281: First time mentioning of "low, baseline, and high LUH2 scenarios" (i.e. maybe not introduced so far?) – it would be nice to have some more information on these three scenarios, also to understand a bit more assumed uncertainties involved with LUH2.

l.342: Please add that the "potential forest or potential non-forest" map described in this subsection is a static map for the whole 850-2100 time-span, or if this is not correct please give more detail.

l. 370: Could you give some information on wood harvest in the different SSPs?

l.501: Why 2010, l.372 indicates start in 2015?

l.507: What does this mean in terms of harvest – is the resulting harvest scenario specific – or taken from one of the two GCAM scenarios? (If the latter, do the two GCAM scenarios differ in terms of wood harvest? If so which one is used?) Why is the GCAM model used and not one of the other IAMs?

l.510: How was pasture treated? As part of grazing-land?

l.517: Can you specify which scenarios did not include slash?

l.591: Could you please clarify if wood harvesting on primary land degrades this to secondary land? (i.e. if harvest on primary land is a transition from primary to secondary land?)– reading on I found this information in line 653

l.654: "whereas wood harvested from secondary land provides an age-(and biomass-) resetting transition" secondary to secondary"." What does this mean? Where is the age and biomass tracked?

l.733: Changes into land use -> agricultural land use?

l.745: Could you add a figure showing the three scenarios? Maybe in the supplementary?

l.862: has -> have

l.910: One of the two bookkeeping models used in the GCBs uses the FAO statistics, the other model - the BLUE model - uses the LUH2 data.

l.913: What do you mean with earlier land cover reconstructions? (not LUH1 since this also includes wood harvest?)

Table 3: Why is there a question mark in "FAO value?" for Fuelwood and Wood harvest?

Table 5 and Table 7: Maybe add the explanation here that negative "Total net transitions" are changes towards agricultural land use?

Fig.4: What is shown in panel b, pasture or total grazing land or? Panel c axis are hardly readable. It is interesting that in comparison to the IAMs LUH2 nearly always seems to have larger crop fractions, do you have an idea why?

Fig.5: Legends are difficult to read.

Fig.8: Category names hardly readable for scenarios. Overlaps of circles and text. Why different names for scenarios and time-periods?

---

## Author Comment (AC1) · 22 Jul 2020

Anonymous Referee #1 () General comments: The authors seek to provide fractional land-use patterns, underlying land-use transitions, and key agricultural management information for the time period 850-2100 at 0.25 degree resolution, all while having a smooth transition between methods applied to past and future results and preserving the future changes. Such an effort is of high importance to both the land surface and climate modelling communities, as maps of this type are essential drivers in simulations studying the feedbacks between the land surface and the climate. This work builds on a previous product, LUH1. Some changes
are an increase in spatial resolution (0.25 degree instead of 0.5 degree); a larger list of 12 subgrid scale land-use types used in the land-use transition matrix; and tracking of key management activities related to agriculture. The authors clearly state assumptions they make in the work at various points in the manuscript, which is fine; I don't require justification for clearly-stated assumptions, although it's always welcome.

Overall the work is high quality, and I believe should be published with only minor revisions.

Specific comments: It was only in section 2.12 that I understood that eight different LUH2 scenarios are available for the future, and (I assume) one for the historical period. I was under the impression that the IAM runs were somehow averaged to create one trajectory from 850-2100. Assuming I am not mistaken, it would be good to clarify that in the abstract and/or introduction, to let users know precisely how many trajectories are available.

> We have now specified the number of scenarios (eight) in the Abstract and the 3rd paragraph of the Introduction.

In section 2, line 125, A and m are solved for. That implies that f and l in Eqs. 1 and 3 are already known. For someone unfamiliar with the process, it's not clear how the inputs listed in lines 127-133 lead to this, before reading all of the detailed description. A short sentence making the explicit link would be welcome here to ease the reader in to the more detailed discussion. For example, when discussing shifting cultivation (line 261), the authors "abandoned the Heinimann et al. (2017) prescribed percentage of total cropland area in the grid cell (e.g. cropland to secondary land), and cleared the same area from natural vegetation". This sounds like it could be A given in Eq 1. Therefore, it's not clear to me what is being solved for in Eqs. 1 and 3, and how the datasets listed in Section 2 contribute to that. Perhaps even a short paragraph would be useful to clear that up. Section 2.11.1 addresses this, but it's easy for the reader to be lost by then.

[Figure]

> Thank you for this constructive feedback. We have attempted to clarify this by simplifying the way in which the cropland management vector is described, and adding a line to the paragraph right before Section 2.1 (which is in addition to the details already given in that paragraph).

The wood harvest numbers are detailed in Section 2.5; however, it's not clear how this feeds into Eqs. 1 and 3. Combined with the forest loss map from 2.6? Perhaps this is described a bit more in Sections 2.10.1 and 2.11.2, but it would be good to keep the reader on track earlier, making sure each section is tied to the overall goal of solving Eqs. 1 and 3.

> We have added a sentence at the beginning of Section 2.5 that references Eq. 1 and points the reader to the additional details that are given later in Sections 2.10 and 2.11.

From Section 2.9, I understand that LUH2 does not simply use output from IAM intercomparison projects or separate runs; significant work was carried out to harmonize the input and create a consistent set of IAM simulations specifically for this project. That deserves more emphasis, in my mind; incorporating a couple words around lines 90 would be good.

> Thank you for this suggestion. We have added several additions to the thrird paragraph of the Introduction to help emphasize this more, including additional references to IAM publications that describe these scenarios in more detail.

Is the paragraph starting around line 525 part of the "Harmonizing inputs" section? It seems to be part of a section dedicated to smoothing the transition between the historical and future datasets, although I don't seen such a dedicated section. I understand "Harmonizing inputs" to be harmonizing the data that goes into the IAM runs. If this paragraph is really discussing that, a sentence to make it more clear would be appreciated. Conversely, if "Harmonizing inputs" is completely dedicated to smoothing the transition between historical and future datasets, I would propose to make that more explicit, as the previous language never connected the idea of "Harmonizing inputs" to

"Smoothing the transition".

> We have modified the title of this section to be "Harmonization of LUH2 Inputs" to make this clearer.

In line 550, no rationale is given for holding the values between 0 to 500. I propose to either explicitly call this an assumption, or to add a sentence/reference giving the reason for these limits (well, the upper limit...limiting fertilizer application to positive numbers makes perfect sense, and thus the lower limit does not need justification). The following sentence could either apply to this choice of 0 to 500 or to the overall method described in the paragraph, and so a couple words to remove ambiguity would be welcome.

> This is a good point. The upper limit of 500 was a simplifying assumption and this has now been made clear in the paragraph about harmonizing fertilizer data inputs.

Figure 1 does a nice job of showing the inputs, model, and outputs. Perhaps a similar chart separated into historical maps, future maps, and the transition would also help the reader?

> Thank you for your appreciation of the model diagram in Figure 1 and for your suggestion to further illustrate the model features by separating this figure into historical maps, future maps, and transitions. However, we intentionally did not provide this separation because an essential feature of our model and dataset is that it harmonizes the historical and future time periods together, while providing a consistent set of states, transitions, and management data. If we were to separate these time periods and data types in the figure, it would imply that they could be considered separately, which is counter to the intended use of this product.

The paragraph beginning in line 707 is quite interesting. As I understand that the eight future trajectories were run specifically for this project, it would be nice to develop this further and see if some of the observations can be explained. For example, the

idea that global cropland area and wood harvesting increased did not surprise me (increased population, increased food supply, wood/bioenergy products increasingly viewed as favorable for the environment). Six scenarios projecting decreased grazing land did surprise me, though, as I would have expected increased meat consumption as more societies shift to the "Western" meat-heavy diet. If the IAM scenarios had been taken from another source, it would not be necessary to explore them further here; however, as it seems that much work went into creating these trajectories for this project, I would appreciate another paragraph dedicated to this. I note that further observations are mentioned in later sections (Sec 3.3, for example), but I don't see any driving factors for the differences.

> Thank you for your interest in these future scenarios. The 8 scenarios were developed as part of ScenarioMIP and are documented in the related publications that we have referenced. We have now added a reference to Popp et al. 2017 and Stehfest et al. 2019 at the beginning of Results section to help clarify this. The beginning of Section 2.8 (future land-use inputs) was also expanded to clarify where these future scenarios come from, and where they are already documented, including a new citation to Riahi et al. 2017.

The badge plots in Figure 8 are great to depict the transition information. I do struggle with the font, though. Could the text indicating the land use (e.g., pasture) be instead put in a legend across the bottom, with the text in the same color as the sector and made larger, at least for the eight trajectories where the font is the smallest (it appears they share the same sectors)?

> Thank you for your appreciation of these figures. We have updated these figures so that they are larger and no longer have the labels for the various land-use types. The colors representing each land-use type are now described in the figure caption.

The authors claim the new LUH2 products contain fifty times the information as the previous LUH1 product on two separate occasions in prominent places (abstract and

conclusion), but do not support this anywhere in the text. It would be best to either add supporting evidence for this claim or remove it.

> We have added additional information to the first paragraph of the Conclusion section that describes the ways in which the data in the LUH2 products contains 50 times more detail.

The evaluation of the LUH2 dataset against diagnostics is most welcome, and it's particularly interesting to see the comparison also against LUH1 for these same diagnostics (Table 3). The diagnostics are not discussed in any detail, though, which makes me wonder how they are "new", as mentioned in the abstract. I would modify the abstract to remove the word "new" or add more discussion on the diagnostics themselves demonstrating that they are, in fact, new, why they were selected, and how tables 3 and 5 compare in their selected diagnostics (they are not identical, for some reason). Additionally, I would be much more interested to read in the abstract what the result of the assessment was rather than just reading that the assessment was done.

> Thank you for your appreciation of our diagnostic assessment. The diagnostics were "new" for LUH2 in the sense that they were significantly expanded from the simple diagnostic assessment reported as part of LUH1. The numerical values reported in the diagnostic tables (Table 3 and Table 5 – which are not identical due to differing time periods) are already cited and discussed in numerous places through-out the manuscript, particularly throughout Section 3.1, so we don't think there is a need to add any additional discussion of them.

Some technical comments: Line 81: "Le Quéré et al. ,2016" should be "Le Quéré et al., 2016"

> Done

Line 771: There is an open paranthesis, but no close in the next line. "reference value of 0.3 X 10 6 km2)."

> Fixed

Line 780: Sometimes petagrams of carbon is written "Pg C" in this paragraph; other-times "PgC". I prefer "Pg C".

> Done. Standardized to "Pg C" through-out manuscript.

Line 791: Four significant figures in these numbers seems pretty optimistic, as that implies an uncertainty of less than 0.1%. I would stick to three at the most (e.g., 13.3%), since this paragraph is reporting differences in the created datasets and not a real uncertainty compared to observations (I'm sure the uncertainty compared to real historical values would be fairly high). Some consistency should be applied to the number of significant digits reported in this paragraph (and likely the whole paper...I see anywhere from two to five for various numbers, with no clear reason).

> Thank you for this suggestion. We have modified the paragraph in question so that the cited values have only 3 significant figures (which is also consistent with the values in the table they are referring to). We have also gone through the whole manuscript and adjusted all numerical values to have no more than 3 significant figures. There are a few exceptions that have 4 significant figures but those are values that we are citing from other publications and datasets, so we have not modified the number of significant figures used in those sources. In some cases we have used fewer than 3 significant figures – this was done when we wanted to improve readability of the data in the tables (for example, by providing all values in a column to the nearest integer), or when the values reported had a much higher known uncertainty.

Line 793: There is a stray "." after 58.

> Fixed

Line 812: The "yr -1" has been split over two lines.

> Fixed

Line 910: Uses "bookeeping". Next line using "bookkeeping". Line 596 uses "book-keeping".

> We have now standardized the manuscript to consistently use the word "bookkeeping", which is consistent with the term used on the Global Carbon Budget annual publications.

---

## Author Comment (AC2) · 22 Jul 2020

Anonymous Referee #2 () In their manuscript "Harmonization of Global Land-Use Change and Management for the Period 850-2100 (LUH2) for CMIP6", Hurtt et al. present the methods used to create the LUH2 dataset – the detailed land cover maps used by the CMIP6 land surface models. The paper is well written and thorough, though it reads like a technical description (which is actually appropriate given the detailed methods required to understand the construction of the LUH2 dataset). I have only minor comments on the manuscript, and believe it is appropriate for publication in GMD.

[Figure]

Line 50 – since the CMIP6 runs have already mostly happened, mentioning that here, and that this is a description/analysis of the land use patterns used by those simulations, would be appropriate (the authors mention it in the discussion).

> Good idea. We have added a line to the abstract that mentions that these simulations have already been performed and that this paper is documenting one of the inputs to those simulations.

Line 64 – typo: provided -> provide

> Fixed

Line 100 – I was confused about what GLM2 was, and what role it plays in LUH2. A bit of elaboration here on that would be helpful.

> We have now clarified this in the first paragraph of the Methods section.

Line 115/116 – drop "i.e."

> Done

Line 120 – (equation 3) Please elaborate more on what equation 3 means. m = f, but the following sentence (lines 121-122) don't sound like m=f...

> Thank you for this feedback. We agree that this equation was somewhat confusing. We have now removed that equation and simplified the way in which we describe the cropland management vector within our system of equations.

Line 155 - At this point, the reviewer wonder's "How are changes in ice and water fraction accounted for?" Are no gridcells that are 100% ice projected to be less than that in the future? Or do those gridcells get to be "bare ground" under the ice, and models with dynamic ice sheet models can convert to the regular land model as appropriate, but just use gravel/bare ground as the ground cover? Or is there a chance for grass to grow on melted glacial sites in 100-300 years? Maybe this dataset assumes land ice is fixed temporally – if so, please state. If not please elborate on how it is handled.

> Section 2.1 already states that "The ice and water fractions of each grid cell were also taken from the HYDE dataset and were assumed constant over time."

Line 203 - Is this because rice is the only C3 crop that is flooded? Is all rice in this dataset represented as flooded?

> Rice is the only crop we consider to be flooded. We do not represent total rice in our dataset – just flooded rice, due to its specific management characteristics. Non-flooded rice is possible but would be part of the remaining C3 annuals and is not represented explicitly, in the same way that wheat is not represented explicitly. We have added a line to Section 2.3 to clarify this.

Line 370 – Section 2.8 provided a very clear and useful walk-through of the land use in each SSP – I thank the authors for laying this out so clearly.

> Thank you

Line 695 - "these 12 states" -> which 12 states? Maybe say "the 12 potential states of vegetation cover used in this dataset..." (remind/help the reader)

> We have now added a line to the beginning of Section 3.1 to clarify this.

Line 728-730 - Elaborate on these three historical scenarios? (It feels a bit like they just showed up - if they were already introduced above it got lost in the details, and reminding the reader of what they are here would be helpful...)

> We have now added a brief description of the 3 historical scenarios in Section 3.1 as well as a sentence introducing them at the very end of the Introduction.

Line 740 – "Badge plots": This is a minor comment, feel free to ignore it. I've never heard the term "badge plot" - I knew what it was when I looked at it, but have always heard of and referred to these plots as "circle network graphs" or "circle network diagrams". Putting "badge plot" into google didn't pop up any figures like this.

> Upon investigation, the official name for these types of figures is "chord diagrams"

and the term "badge plots" has now been replaced throughout the manuscript with the term "chord diagrams".

Line 760-763 – This is quite interesting! At this point, I wanted to hear more about it. The authors actually discuss it more later.

> Thank you

Line 785 – Typo: "above- ground" -> "above-ground" (remove space after dash)

> Fixed

Line 788-789 – as mentioned above, it wasn't entirely clear what the lower, baseline, and upper historical scenarios were.

> As described above we have now added a brief description of the 3 historical scenarios in Section 3.1 as well as a sentence introducing them at the very end of the Introduction.

Line 814 – Is this the same metric as used on Line 760? (If so, consider moving the two discussions to the same place - when I hit line 760 I wanted to know more about what it used to look like.)

> These metrics both deal with secondary mean age, however on line 760 we are discussing the globally averaged secondary mean age in years 2000 and 2015, whereas on line 814 we are discussing the range of secondary mean age values in the 1900s across all regions/continents. We have separated these two discussions into two different sections to highlight the global nature of our results (which are compared with diagnostic reference values), as well as the spatial-temporal nature of the underlying data.

Line 824 - tied to the above comment - secondary forest age is going up by quite a bit in RCP 8.5?

> Although RCP8.5 has the highest secondary mean age, these values did not vary

significantly across scenarios, as mentioned in the manuscript. Line 936 - This is the list of datasets for Ma et al 2019 - maybe it is the same list, in which case no modification is necessary. If there is a separate list for this manuscript, then this link needs to be updated.

> Thank you for pointing this out. We have updated this section to refer to the actual GLM2 code archive used to generate the LUH2 datasets.

Figure 4 – Why the discrete drop in pasture-land after 2000? Cropland has a smoother transition from historical to projected land area.

> Pasture is typically more challenging to define consistently across models. We have now added mention of this when we discuss Figure 4 in Section 2.9.

Figure 5 – text is extremely small in subplot legends. Had to blow it up on the computer, and wouldn't be able to read it in print.

> These figures have now been updated, with the legends removed and described in the figure caption instead.

Figure 8 – as in Figure 5, text is too small to read

> We have updated these figures so that they are larger and no longer have the labels for the various land-use types. The colors representing each land-use type are now described in the figure caption.

Figure 9-14 – are these images blurry?

> Thank you for pointing this out. We have updated these images to improve their resolution, and they look much better now.

Figure 15 – legend text too tiny to read

> Thank you for pointing this out. We have updated these figures to increase the font size in the legends and the standardize the labeling.

---

## Author Comment (AC3) · 22 Jul 2020

Short Comment #1 The paper by Hurtt et al. describes the LUH2 dataset and the harmonisation required to achieve a continuous dataset of human land-use activities required for CMIP6 ESM simulations. Thank you for this tremendous effort and this dataset.

While the scope of the methods and the results seem to cover the important aspects, I would appreciate if the discussion section would be a bit more elaborate. Topics that could be helpful to be discussed would be (a) in particular a discussion of emerged/possible issues for the usage of the data in LSMs/ESMs, since this is stated

to be one of the main goals of the dataset (three examples coming into my mind are (1) how to deal with the interpolations used to construct wood harvest; (2) how to deal with inconsistencies between the static forest/non-forest-map used in LUH2 when calculating with another (disagreeing) vegetation cover; (3)how to treat rangelands);

(b) a discussion of uncertainties (including the three scenarios but if possible even broader; further examples could be pointing to where uncertainties of underlying data are discussed; conflicts of assumptions taken with shifts in country borders; climate dependent changes in biomass (or is this accounted for in the MIAMI-LU model?),...);

(c) comparisons to other datasets, e.g. the land cover dataset ESA-CCI?

E.g. regarding (1) the interpolations used to construct wood harvest: the two time dependent interpolation rules (regarding slash fractions and inclusiveness of wood cut in conversions in the wood harvest statistics) probably cause problems for a straightforward usage of the data in ESMs. The fate of the carbon in the LSMs matter, i.e. it is important to which carbon pool the harvested carbon should be assigned (such as construction wood, fire wood, etc, since these have different turnover times).

(2) albeit LUH2 is a land-use data set and not a land cover dataset, there are several assumptions where information on the land cover is used in the construction of the LUH2 data and which might be in conflict with the land cover assumed in an ESM simulation?

(3) there has been quite some discussion about this in the TRENDY process, particularly if rangelands involve land cover change, or not. Furthermore, while reading I sometimes had the feeling that knowledge required for understanding was only given later (e.g. how grazing lands are treated and when rangelands and pastures are aggregated for grazing lands and information about the uncertainty scenarios).

> Thank you for these constructive thoughts and suggestions. The manuscript is already quite lengthy and, as mentioned, there is already a discussion of many of these

issues in other publications (that we have cited). However, we have added a couple of additional paragraphs to the Discussion section to address issues 2 and 3 above as well as comparisons to other datasets.

Some specific comments: l.80: Starting the 2016 version of the global carbon project (GCB) LUH2 has been used, i.e. LUH1 has not been used for the following GCBs: Le Quéré et al., 2016; Le Quéré et al., 2017; Le Quéré et al., 2018; Friedlingstein et al., 2019

> Modified around line 80 and line 925

l.281: First time mentioning of "low, baseline, and high LUH2 scenarios" (i.e. maybe not introduced so far?) – it would be nice to have some more information on these three scenarios, also to understand a bit more assumed uncertainties involved with LUH2.

> Thanks for this feedback. We have now added a brief description of the 3 historical scenarios in Section 3.1 as well as a sentence introducing them at the very end of the Introduction.

l.342: Please add that the "potential forest or potential non-forest" map described in this subsection is a static map for the whole 850-2100 time-span, or if this is not correct please give more detail.

> Done

l. 370: Could you give some information on wood harvest in the different SSPs?

> The various scenarios (including wood harvest) are well documented in the cited publications. We have added an additional reference to this section to provide additional information.

l.501: Why 2010, l.372 indicates start in 2015?

> IAM simulations began in 2010 and data is reported from that date onwards, even though the harmonization between historical reconstructions and future scenarios occurs in the year 2015.

l.507: What does this mean in terms of harvest – is the resulting harvest scenario specific – or taken from one of the two GCAM scenarios? (If the latter, do the two GCAM scenarios differ in terms of wood harvest? If so which one is used?) Why is the GCAM model used and not one of the other IAMs?

> We have added a clarification at the end of paragraph 1 in Section 2.9 that the wood harvest comes from the analogous scenarios computed by the GCAM model.

l.510: How was pasture treated? As part of grazing-land?

> Managed pasture is part of grazing land (along with rangeland). We have now defined this in paragraph 2 of Section 2.9.

l.517: Can you specify which scenarios did not include slash?

> No IAM scenarios included slash, so we have modified this sentence to remove the phrase "to those that did not include slash", since we added it all scenarios.

l.591: Could you please clarify if wood harvesting on primary land degrades this to secondary land? (i.e. if harvest on primary land is a transition from primary to secondary land?)– reading on I found this information in line 653

> Since this information was provided and found on Line 653, we assume nothing else needs to be added here.

l.654: "whereas wood harvested from secondary land provides an age-(and biomass-) resetting transition" secondary to secondary"." What does this mean? Where is the age and biomass tracked?

> We have added a clarification in Section 2.11.2 that describes the variables that track these values and that the "resetting transition" represents a reduction in age and biomass.

l.733: Changes into land use -> agricultural land use?

> Since urban land is also included in the LUH2 dataset, the changes into land use do not only include agricultural land.

l.745: Could you add a figure showing the three scenarios? Maybe in the supplementary?

> Thank you for your interest in viewing additional figures related to these scenarios. Although we would like to show as many figures as possible to represent this dataset, the paper is already pushing the limits of length, and there are many possible additional figures that could be included due to the richness of the dataset. The data always freely available for download and use if readers would like to view their own analysis of specific parts of the dataset.

l.862: has -> have

> Fixed

l.910: One of the two bookkeeping models used in the GCBs uses the FAO statistics, the other model - the BLUE model - uses the LUH2 data.

> Thank you for pointing out this error. We have now corrected it in the Discussion section, where the two bookkeeping models are mentioned.

l.913: What do you mean with earlier land cover reconstructions? (not LUH1 since this also includes wood harvest?)

> We agree that this sentence was confusing, and did not helpfully add to the discussion, so we have removed it.

Table 3: Why is there a question mark in "FAO value?" for Fuelwood and Wood harvest?

> Thank you for catching this error. We have now added the actual FAO reference values to the table.

Table 5 and Table 7: Maybe add the explanation here that negative "Total net transitions" are changes towards agricultural land use?

> Negative net transitions describe net transitions away from land-use (not towards land use). The definitions of net (and gross) transitions are already given in Section 3.1 so we don't think they need to be repeated in the table caption.

Fig.4: What is shown in panel b, pasture or total grazing land or? Panel c axis are hardly readable. It is interesting that in comparison to the IAMs LUH2 nearly always seems to have larger crop fractions, do you have an idea why?

> We fixed the inconsistency between "pasture" and "grazing land" in the figure title and caption. We also made the axis labels more readable. Although the LUH2 data does appear to have larger crop fractions than the IAM data in some scatter figures, this is not true for all scenarios and varies from model to model based on the assumptions each IAM makes (which the density of black dots can sometimes hide).

Fig.5: Legends are difficult to read.

> We updated the legends to improve this figure.

Fig.8: Category names hardly readable for scenarios. Overlaps of circles and text. Why different names for scenarios and time-periods?

> We have updated these figures so that they are larger and no longer have the labels for the various land-use types (and do not overlap each other). The colors representing each land-use type are now consistently described in the figure caption.